# AlphaZero Neural Scaling and Zipf's Law: a Tale of Board Games and Power Laws

**Oren Neumann, Claudius Gros**
Institute for Theoretical Physics
Goethe University Frankfurt
`oneumann.papers@gmail.com, gros@itp.uni-frankfurt.de`

## Abstract

Neural scaling laws are observed in a range of domains, to date with no universal understanding of why they occur. Recent theories suggest that loss power laws arise from Zipf's law, a power law observed in domains like natural language. One theory suggests that language scaling laws emerge when Zipf-distributed task quanta are learned in descending order of frequency. In this paper we examine power-law scaling in AlphaZero, a reinforcement learning algorithm, using a model of language-model scaling. We find that game states in training and inference data scale with Zipf's law, which is known to arise from the tree structure of the environment, and examine the correlation between scaling-law and Zipf's-law exponents. In agreement with the quanta scaling model, we find that agents optimize state loss in descending order of frequency, even though this order scales inversely with modelling complexity. We also find that inverse scaling, the failure of models to improve with size, is correlated with unusual Zipf curves where end-game states are among the most frequent states. We show evidence that larger models shift their focus to these less-important states, sacrificing their understanding of important early-game states.

## 1   Introduction

Neural scaling laws, describing the scaling of model performance with training resources, have been documented across many architectures and use cases [1, 2, 3, 4, 5]. The performance of these models, typically measured by test loss, follows a power law in either compute, model size or training data. The robustness of these power laws and their usefulness in training large models, especially large language models (LLMs) [6], prompted a range of attempts to find a model explaining the mechanism behind neural power-law scaling [7, 8, 9, 10, 11]. Several models connect performance power laws to Zipf's law, a universal power law appearing in natural language [12, 9, 13, 14, 15, 16].

Scaling laws in reinforcement learning (RL) are so far relatively rare compared to supervised learning [17, 18, 19]. AlphaZero, a multi-agent RL algorithm that beat human champions in games like chess and Go [20], exhibits power laws of Elo score with training resources, as well as a power law relation between compute and optimal model size [5, 21, 22].

In this paper, we explore the mechanism behind known RL scaling laws, comparing this mechanism to the quantization model of LLM scaling [12]. Our main findings are:

1. We find that AlphaZero agents that exhibit scaling laws, such as those trained in Neumann & Gros (2022) [5], produce train and test data that follows Zipf's law, suggesting that learned tasks also scale with Zipf's law.

2. We show how these smooth Zipf laws form when agent policies are combined with the tree-structure of board games, which is known to create a Zipf-like state distribution [23].

39th Conference on Neural Information Processing Systems (NeurIPS 2025).

We find a correlation between the Zipf curve and scaling exponents, exposed by modulating policy temperature at inference.

3. In agreement with the quanta scaling model, we show that agents minimize loss on game states in descending order of frequency, counter-intuitively achieving *better performance on harder tasks.*

4. *Inverse scaling,* the abrupt failure of scaling laws at large sizes, coincides with an unusual state-frequency distribution. We show that large models scale negatively on games with an unusual tree structure that increases the frequency of strategically-unimportant late-game states. Larger agents achieve better loss on these late-game states, sacrificing performance on more important early-game states. We propose an explanation of inverse scaling in AlphaZero using a combination of the quantization model and the selective non-stationarity of training targets.

All code and data required to reproduce our results is publicly available[1].

## 2   Background

A short description of the AlphaZero algorithm and AlphaZero scaling laws is given in appendix A.

**Zipf's law**   Zipf's law is an empirical law describing the distribution of values when sorted in decreasing order, typically with respect to frequency [24]. When sorting elements in decreasing order of frequency, e.g. words in natural language texts, one regularly finds that the frequency distribution $S(n)$ follows a power law in element rank $n$:

$$S(n) \propto \frac{1}{n^\alpha} \,, \tag{1}$$

often with $\alpha \approx 1$ for natural language datasets [25, 26].

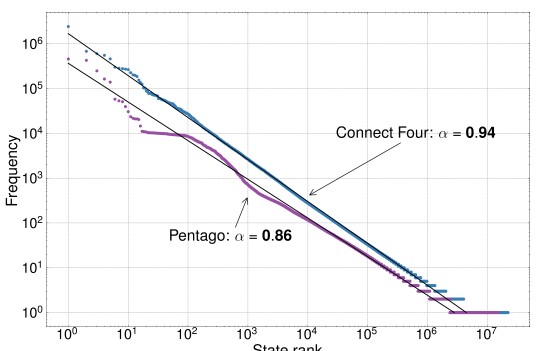

Figure 1: **Zipf's law in AlphaZero games.** Board-state frequency follows a power law in state rank, here for Connect Four and Pentago. Similar exponents $\alpha$ appear for agents of various sizes trained on different games, see appendix C.1.

**Quantization model of LLM scaling** Michaud et al. (2023) [12] proposed that a 'quantization model' may explain the origin of power-law scaling in LLMs, connecting neural-network training to language Zipf's laws. The model assumes that LLMs learn discrete task quanta and that the frequency of these tasks follows Zipf's law. The quantization model suggests that language models learn tasks *in descending order of frequency*, which leads to power-law scaling laws. If the loss on each task is reduced by a fixed amount $\Delta L$ after it is learned, then a model that learned the first $n$ tasks will have an expected loss of

$$L = \frac{\Delta L}{\alpha\zeta(\alpha+1)} \cdot n^{1-\alpha} + L_\infty \implies (L - L_\infty) \propto n^{1-\alpha} \,, \tag{2}$$

where $\zeta$ is the Riemann zeta function. It is assumed that a fixed neural capacity $c$ is needed to fit each of the independent task quanta. At the limit of infinite compute and data, the number of learned task quanta $n = \frac{N}{c}$ is determined by the number of parameters $N$. Eq. 2 then becomes a power-law scaling law:

$$(L(N) - L_\infty) \propto \frac{1}{N^{\alpha-1}} \,, \tag{3}$$

and the Zipf exponent $\alpha$ determines the size-scaling exponent $\alpha_N = \alpha - 1$. Similar reasoning relates compute- and data-scaling laws to Zipf's law.

---

[1]Code and data available at github.com/OrenNeumann/alphazero_zipfs_law.

**Other supervised learning scaling models** A large body of work exists on models of neural scaling laws on static datasets, several of which making the connection between Zipf's law and scaling laws. Hutter (2021) [9] argues that data scaling laws emerge when training on Zipf-distributed data under the assumption of infinite memory. Bordelon et al. (2020) [13] develop a model of data scaling laws for kernels, which applies to neural nets at the infinite width limit. Other works expand this line of work on kernel models, including results on real datasets [14, 27]. Maloney et al. (2022) [15] develop a model that explains joint scaling of model size and dataset size. Cabannes et al. (2023) [16] assume Zipf-distributed data and develop a model for scaling of attention models. Schaeffer et al. (2025) [28] propose that exponential scaling on individual tasks can aggregate to a power law when the distribution of success probabilities is heavy tailed, in the case of multiple attempts per task. Dohmatob et al. (2024) [29] explore model collapse through the effect of synthetic data on heavy-tailed data distributions.

We frequently compare our results to Michaud et al. (2023) [12] throughout this paper, since their model develops the full set of scaling laws for neural nets (compute, parameters, data) by assuming a Zipf's law of the data. Our comparisons also hold for the above-mentioned works, depending on the assumptions they made. To our knowledge, no complete model exists that explains scaling laws in any RL setting.

## 3 Related work

**Finding Zipf's law in board games** To our knowledge, no study has been made on Zipf's law in AI games. Blasius & Tönjes (2009) [23] and Georgeot & Giraud (2012) [30] show that the popularity of openings in human game datasets follows Zipf's law in chess and Go, respectively. Blasius & Tönjes are the first to propose a connection to the branching-tree structure of chess, without proof. We provide evidence supporting such a connection in section 5.

**Analyzing concept learning in AlphaZero** In sections 7 and 8 we present results on the scaling of AlphaZero loss on individual states. A similar analysis is done by McGrath et al. (2022) [31] and Lovering et al. (2022) [32], who both analyze how human concepts are learned by AlphaZero by measuring accuracy. McGrath et al. (2022) [31] map how human chess concepts are learned, both as a function of training length and neural network depth. They measure the accuracy of a few hand-crafted concepts, in contrast to our analysis that measures loss on up to $10^5$ unique states. Similar to our inverse-scaling results in section 8, Lovering et al(2022) [32] observe differences between how short-term and long-term concepts are learned in the game of Hex.

**Effects of state frequency on training** Our work discusses the effects of training-data state frequency on AlphaZero agent performance. Another observation regarding the relation of data frequency with performance can be found in Ruoss et al. (2024), [33], where Transformers are trained with supervised learning on annotated human chess games. The authors perform an ablation study where they change the frequencies of board states in the dataset, either sampling states uniformly or sampling them according to their frequency in human games, which follows Zipf's law. Testing on chess puzzle accuracy, they find that changing the state frequency has a strong effect on model performance, favoring uniform sampling over sampling from the Zipf distribution.

## 4 Methods

We use AlphaZero agents trained on four board games: Connect Four, Pentago, Oware and Checkers. All neural nets are either open-source models trained in Neumann & Gros (2022) [5], or newly trained using the same open-source OpenSpiel code [34]. We use the open-source model weights for Connect Four and Pentago, and train new models with the same hyperparameters on Oware and Checkers. All of these models exhibit power-law scaling laws, either partially or fully: Connect Four and Pentago scaling laws are discussed at length at Neumann & Gros [5], here we present size-scaling curves for Oware and Checkers, see Fig. 5**A**. The power laws are for the Bradley-Terry playing strength $\gamma$, which determines the Elo score: Elo $\propto \log_{10}(\gamma)$ [35, 36]. One finds that $\gamma$ scales as a power of neural-net parameters and training compute [5]:

$$\gamma \propto N^{\alpha_N} , \gamma \propto C^{\alpha_C}. \tag{4}$$

Most results in this paper involve the frequency of states $s \in \mathcal{S}$ in a Markov decision process $\mathcal{M} = \langle \mathcal{S}, \mathcal{A}, \mathcal{P}, \mathcal{R} \rangle$ [37]. In practice, we identify each state by its observation tensor, which is the

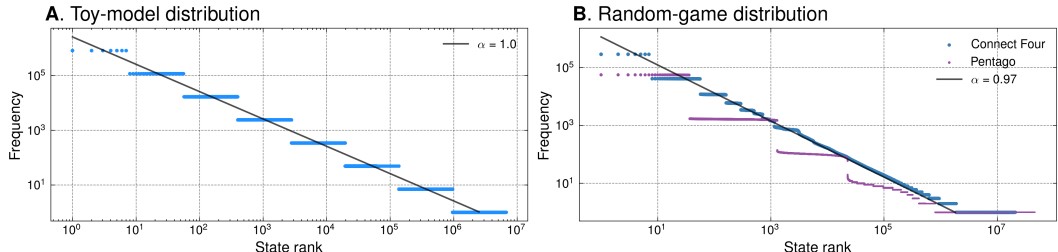

Figure 2: **Zipf power laws emerging from the tree structure of board games. A:** In a toy-model game where board states branch out symmetrically, state frequency is organized as a series of plateaus centered around a linear line. **B:** Adding game rules, but still playing random moves, the plateaus are smoothed out, producing a power law with a slightly different exponent. This is in particular evident for Connect Four, which has a much lower branching factor than Pentago. The tail plateaus are caused by the finite amount of data.

input to the neural net. In all four games, this observation is the position of each piece on the board, and does not include the history of previous moves (unlike the chess analysis of Blasius & Tönjes (2009) [23]). All AlphaZero agents use MLP nets with 3 hidden layers at exponentially-increasing widths, with a fixed amount of 300 MCTS steps per move during both training and inference.

## 5 Zipf's law in AlphaZero

**AlphaZero board state distribution**    We find a Zipf distribution in games played by AlphaZero agents that exhibit scaling laws. Fig. 1 shows the frequency distribution of board states played by Connect Four and Pentago AlphaZero agents of varying sizes during training. The number of times each state was visited follows a clear Zipf power law when sorted in descending order. The exact value of the exponent defined in Eq. 1 fluctuates in the range $\alpha \sim 0.8-1.0$ for different models and games, close to the value observed in human Go games [30] and to the exponent of random games, discussed in the next section. Appendix C.1 contains plots of individual-agent Zipf curves, and discusses the variation of the power-law exponent $\alpha$. The plateaus at the tail-end of each distribution are due to finite-size effects, and smooth-out when more state-frequency data is available.

This result strengthens the assumption that Zipf's law is not caused by human decision-making, but is rather derived from the game rules. The existence of Zipf's law in *human* games has long since been established, but its cause is not entirely clear. The frequency of popular opening moves in games like Chess and Go follows a descending power law in rank, both for expert and amateur human players [23, 30]. The fact that an RL model trained with no human knowledge produces Zipf's law similar to humans implies that Zipf's law is not caused by human strategy preference.

**Visualizing the origin of Zipf's law**    A first intuition regarding the source of Zipf's law is presented in Fig. 2. The relation of Zipf's law to the underlying branching process of board games was already pointed out by Blasius & Tönjes (2009) [23]. It is easy to show that an ideal game, with a constant branching factor and a tree structure free of loops, would produce a Zipf-like distribution. Specifically, one can show that the frequency distribution in such a game is a series of plateaus around the line $S(n) \propto n$, see appendix J. The same reasoning explains the appearance of Zipf's law in randomly-generated texts [38]. In Fig. 2**A** this ideal-game Zipf's law is visualized by plotting the state-frequency distribution for a non-looping, constant branching game with fixed length. Increasing the branching factor widens the plateaus, spreading them out. We use the initial branching factor of Connect Four.

In Fig. 2**B** we show that adding game rules molds the plateaus into a smoother curve. Using a uniformly random policy to play Connect Four and Pentago smooths the power-law-centered plateaus, roughly merging them into a straight line. The toy-model exponent changes slightly after adding game rules, but the general power-law trend remains intact.

Appendix C.2 contains additional results on the emergence of Zipf's law, and shows that non-uniform policies can also smooth the distribution into a power law. As we see in Fig. 1, a learned policy smooths the distribution further, keeping the power-law shape while slightly changing the exponent.

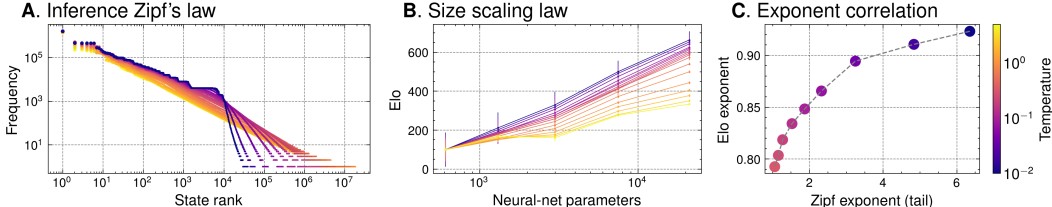

Figure 3: **Measuring correlation between Zipf's law and scaling exponents.** By changing the move-selection temperature at inference time, both the state-distribution Zipf's law and the size-scaling law are augmented, plotted here for Connect Four. **A:** As $T \to 0$, the Zipf curve starts to bend, following a steeper power law at high-ranks. **B:** $T$ also changes the Connect Four size scaling law[2], either directly by lowering policy quality at high $T$, or indirectly by changing the game state distribution. **C:** By modulating $T$ at low values, we plot the dependence of the scaling power law on Zipf's law, using the tail exponent.

## 6  Temperature and correlation with scaling laws

Although measuring the correlation between Zipf's law and scaling laws is not straightforward, we present how this can be done using temperature. Natural language Zipf exponents are impossible to change in real-world data. In contrast, RL Zipf laws can be augmented by changing the policy temperature $T$. AlphaZero uses Monte-Carlo tree search (MCTS) to generate a policy $\pi$:

$$\pi(s_0, a) = \frac{N(s_0, a)^{1/T}}{\sum_b N(s_0, b)^{1/T}} \tag{5}$$

where $N(s, a)$ is the number of times the state-action pair $(s, a)$ was probed, see appendix F for a detailed explanation. The temperature $T$ interpolates between a deterministic policy choosing the best move at $T = 0$, and a uniform policy at $T = \infty$. At inference time, e.g. when calculating Elo ratings, $T$ is either set to zero or reduced to a low value, the latter to avoid deterministic policies.

Temperature has a strong influence on the state-frequency curve, as we show for Connect Four games in Fig. 3**A**. As $T$ approaches zero, the Zipf curve starts to bend due to the formation of plateaus of popular games that are played repeatedly when temperature is low. Deviations from these games make up the post-bend tail. The Zipf exponent of the tail is not well-defined for $T \to 0$ for a finite number of agents and finite numbers of games played. The bending of the frequency curve with decreasing $T$ is somewhat intuitive, considering how temperature affects games. At low $T$, fewer unique states are played, meaning the curve must end at a lower rank. This pushes up the rest of the frequencies, since the total number of states played remains constant.

**Exponent correlation**  In Fig. 3**C** we tune the inference temperature $T$ to analyze the correlation between the size scaling law (Fig. 3**B**) and the Zipf exponent (Fig. 3**A**). The two exponents change together, indicating that the inference-time state distribution does affect the scaling law, as predicted by the quantization model. Unlike the linear correlation predicted by the quantization model for LLMs, we observe a non-linear relation. The non-linearity is expected when one considers the $T \to 0$ limit, in which RL scaling laws converge to a finite exponent while the Zipf tail exponent diverges to infinity.

Additional experiments confirm that the relation between the state distribution and the scaling law is causal, meaning the scaling law changes only because of the changing Zipf distribution. A possible non-causal effect could be Elo scores changing due to the degradation of the policy $\pi$ at high $T$. We measure the effect of $T$ on policy quality in appendix D and find that the probability the policy $\pi(T)$ assigns to optimal moves only starts decreasing at $T > 0.5$. The correlation in Fig. 3**C** is strictly causal, since it falls in the temperature range where policy quality is preserved. Interestingly, the Zipf exponent only changes in the temperature where policy quality is optimal, and is fixed for $T > 0.5$.

**Limitations**  Augmenting policy temperature is not a perfect comparison method, since it is not clear how much influence low-rank plateaus have on agent performance. This is especially true at the

---

[2]Where we plot Elo scores (Figs. 3,5), we set the Elo of the lowest agent to 100. Each curve in Fig. 3**B** is to scale, but the positions of the curves are not to scale.

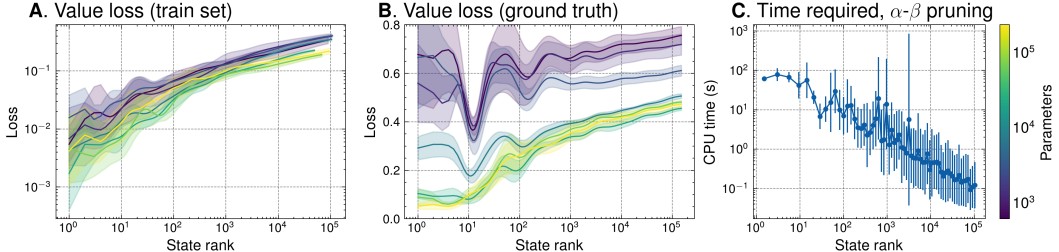

Figure 4: **Connect Four value-loss[4] scaling with rank. A:** Average loss on each agent's own training data. Loss steadily increases with rank. Larger agents achieve better loss. **B:** Loss on a dataset annotated by a game solver. Larger models are closer to the ground truth values of optimal play. The overall trend of loss increasing with rank is surprising, when one considers that the complexity of states *decreases* with rank. See appendix E for detail on error bars. **C:** The time needed to evaluate the same states using alpha-beta pruning [39] drops on an exponential scale with rank (mean and standard deviation are geometric).

lowest temperatures ($T < 5 \cdot 10^{-2}$), where the plateaus are most dominant, accounting for $75 - 90\%$ of all states visited. In this temperature range, it is hard to tell how influential is the tail distribution compared to the plateaus.

Temperature can augment the Zipf curve since it affects exploration, which in turn affects the state distribution, as we also show in appendix C.2. This implies that changing the MCTS parameters could also have an effect on state distribution, but we did not verify this relation.

## 7    Analyzing how loss scales with Zipf's law

In agreement with the quantization model [12], we find that AlphaZero models reduce their loss on game states *in descending order of frequency*. In Fig. 4 we plot the value-head loss on Connect Four training data in the limit of abundant compute, tracking the loss of each agent on its self-play games.[3]

**Comparison to the quantization model**    The quantization hypothesis assumes models learn tasks in a binary way, sharply reducing the loss on each task in descending order of frequency. As a function of neural net size $N$, this assumed loss function is a step function that changes from low to high loss at rank $n \propto N$ [12]. In comparison, we find that AlphaZero loss increases smoothly with rank rather than resembling a step function, likely because board positions are not independent task quanta.

In Fig. 4**B** we plot the value loss $L_{\text{value}} = (z - v)^2$ on ground-truth labels $z$, calculated by an alpha-beta pruning solver [39]. We see that larger agents have significantly better ground-truth loss over smaller agents, confirming that increasing model size improves the absolute value loss. Ground-truth value loss is directly related to better performance, similar to test loss in LLMs that is known to correlate with accuracy on downstream tasks [40, 41, 6].

**States are optimized in descending order of frequency & complexity**    It is surprising that absolute value loss (Fig. 4**B**) tends to increase with state rank, since higher rank states are easier to model, meaning loss *increases* as state complexity *decreases*. Low-loss states are the hardest to model using conventional methods, as we show in Fig. 4**C** for alpha-beta pruning [42]. The most frequent states have a larger state tree branching out from them, making it harder for search-based algorithms like MCTS to tackle them. As a result, the CPU time required to calculate the ground truth value of a state decreases steadily with rank on an exponential scale.

---

[3]The largest nets approach the optimal play limit [5], causing loss to stagnate as model size increases to the largest values.

[4]Note that AlphaZero's value head is not equivalent to the value net of actor-critic methods, see appendix F.

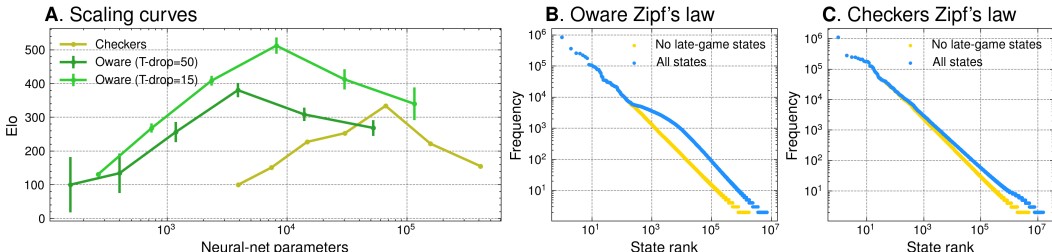

Figure 5: **Inverse scaling in AlphaZero. A:** Agents playing Checkers and Oware follow a size scaling law that abruptly changes direction, when large models fail to utilize their capacity. The scaling curve does not flip due to approaching the perfect-play ceiling: training a suite of Oware agents with different hyperparameters (light green), they extend to higher Elo scores. Error bars are one standard deviation. **B:** Oware games played by AlphaZero follow Zipf's law, interrupted by a small plateau. This is caused by high-frequency late-game states, present due to the unusual tree-structure of Oware, see Fig. 6. **C:** Checkers shares the same tree structure, but the effect on the Zipf curve is less visible.

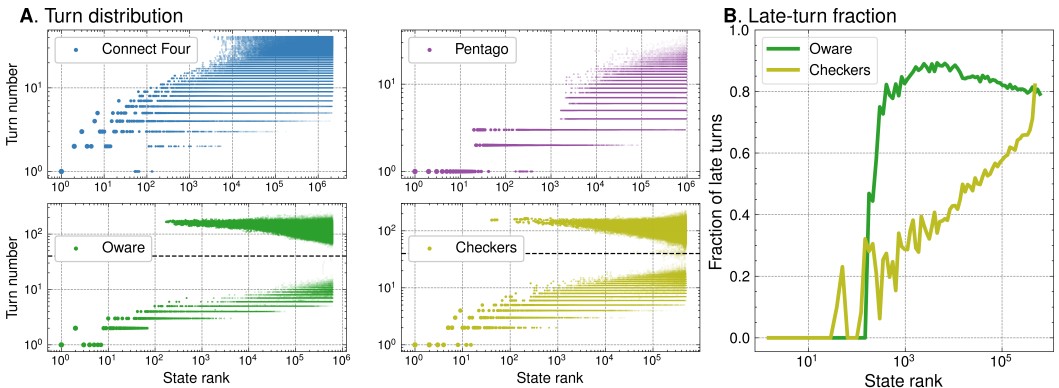

Figure 6: **Turn distribution anomaly coincides with inverse scaling. A:** When the game tree spreads exponentially, e.g. in Connect Four and Pentago, turn number correlates with state frequency. However, in games with inverse scaling (Oware and Checkers) the game tree converges to a comparatively small set of endings, resulting in high-frequency late-game states. **B:** The fraction of late-game states (above dashed line in Oware and Checkers) rises either sharply or gradually with rank.

# 8 Connecting inverse scaling to the game structure

Inverse scaling, i.e. performance scaling negatively with training resources, is known to occur in LLM training [43]. RL models can similarly fail to scale reliably, sometimes decreasing in performance as neural-network size and compute budget are increased. We see a clear example of inverse scaling for AlphaZero agents trained on Checkers and Oware, as shown in Fig. 5A. These agents follow a steady scaling curve that abruptly breaks down beyond a certain neural network size, where Elo starts to scale negatively with size.

## 8.1 Inverse scaling coincides with a frequency-distribution anomaly

As we show in Fig. 5B, the state-frequency distribution of Oware games is visibly different from the Zipf's law present in other games. The Oware Zipf curve is interrupted by a bump starting after rank 100, caused by a group of states with roughly constant frequency. The Zipf power-law exponent remains unchanged before and after this interruption. In this section, we show that the anomalous Zipf curve is caused by a property of the Oware game tree. The same anomaly occurs in Checkers, but is harder to detect solely by looking at frequency curves.

**Turn distribution anomaly**   The distribution of state turn numbers exposes a distinct difference between games with clean scaling laws and games with inverse scaling. In Fig. 6 we plot the average number of moves played in an episode before encountering each state. Connect Four and Pentago turn numbers are strongly correlated with state rank, due to the number of possible states diverging exponentially with time. In contrast, Oware and Checkers display forked turn distributions where a substantial fraction of the most frequent states appear late in the game. In Oware, the sharp change in the fraction of late states (Fig. 6**B**) happens at the same rank where the Zipf curve bump appears.

States above and below the dashed line in Fig. 6**A** are clearly distinguishable when visualized, see examples in appendix G. Late-game states above the dashed line are positions close to the end of a game, when the board is mostly empty, while early-game states below the line are opening moves, i.e. small variations of the initial board state.

**Anomaly is caused by game rules**   The prevalence of high-frequency late-game states in Oware and Checkers data is a direct result of the game rules, which shape the game-tree structure. In both games, pieces are successively removed from the board, ending the game when one of the players has no pieces left (infinite-loop draws are rare). This win condition produces game trees that expand and then contract: starting from a predetermined position, expanding exponentially every move, but then contracting again towards the end of the game as the number of pieces decreases and with it the number of possible board-state permutations. The probability that a game ends in a frequently-visited board state is high.

In contrast, Connect Four and Pentago have a different tree structure. These are line-completion games where pieces are repeatedly added to an empty board, and the number of possible full-board combinations is still high. Go and Chess also do not have converging game trees; a Go game typically ends with many pieces on the board, with large combinatorial complexity. Chess is a piece-elimination game similar to Checkers, but winning is not conditioned on capturing all the opponent's pieces and checkmates can occur while the board is relatively full.

## 8.2   Relation to inverse scaling

The occurrence of inverse scaling at large model sizes as well as an anomaly in the train-set distributions suggests that the two phenomena might be linked. A possible connection is that an increased neural capacity leads to higher-rank states having more influence on model training. Here we present evidence for such a connection between state distribution and inverse scaling.

**Oware loss spikes at transition point.**   As we show in Fig. 6, late-game states (above the dashed line) dominate the Oware distribution at higher ranks, in a sharp transition from $0\%$ to $> 80\%$ of all states. This transition is drastic enough to visibly skew the frequency distribution seen in Fig. 5**B**, bending the Zipf power law. The distribution shift causes Oware value loss to spike, as we show in Fig. 7**A**. The loss of all but the largest models jumps by up to an order of magnitude at the transition point, showing that models that fit early-game states well have a hard time fitting late-game states. This counter-intuitive behavior of getting worse loss on easier states, which we discussed in section 7, is even more surprising here: Oware states after the transition point are predominantly end-game configurations, and should be *significantly* easier to model than early-game states using search algorithms.

The quantization model provides an explanation for the sharp rise in loss. As models fit the most frequent states, smaller models do not have enough capacity to additionally model less frequent states accurately. The large qualitative difference between early- and late-game states makes the difference in loss especially pronounced. Interestingly, larger agents converge on a smooth loss curve similar to that of Connect Four, at the cost of higher loss on early-game states.

**Large models overfit on late-game states.**   Splitting the loss curve to early- and late-game states paints a clearer picture. Loss on early-game positions scales smoothly for all models (Fig. 7**B**), with smaller models achieving lower loss than larger models. Surprisingly, large models have better loss on late game states (Fig. 7**C**) following a reversed order compared to early-game states. We note that the data is collected from games at late stages of training, when game quality is high and relatively constant.

Inverse scaling is likely caused by large model overfitting on late-game states. Larger models fit end-game states better, for the price of worse performance on game openings. Opening moves are

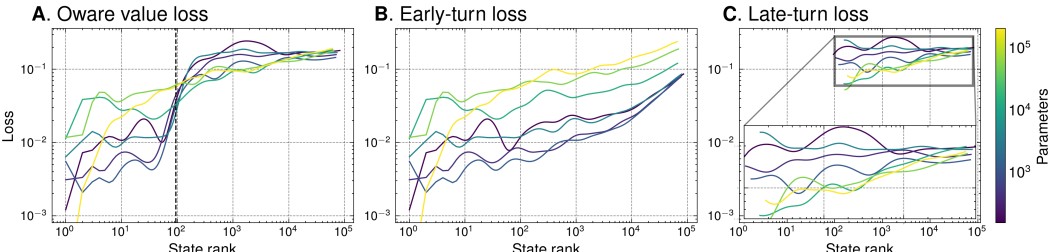

Figure 7: **Increasing neural capacity leads to overfitting on end-game states in Oware. A:** Value training loss sharply increases at the point where late-game moves dominate the state distribution (dashed line). Late-game states have much higher loss despite being much easier to model. **B&C:** Looking at the loss on late- and early-game states separately, we see that larger models predict late-game states progressively better, but predict less-frequent, early-game states progressively worse. Increasing the capacity allows larger models to optimize intermediate-rank states, shifting their focus to frequently appearing late-game data. Forgetting crucial early-game information could explain the onset of inverse scaling in Oware.

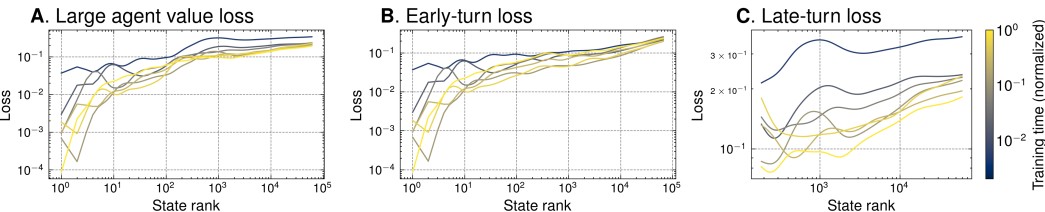

Figure 8: **Large models forget knowledge on early-game states. A:** The loss of the largest Oware agent improves (on average) during training. **B:** On early-game states, loss initially goes down, then stops improving and eventually worsens towards the end of training. **C:** Loss on late-game states steadily improves during training. The fact that large nets fit early-game states but then forget them suggests that target non-stationarity might be the cause of inverse scaling.

crucial for good performance, while end-game states are mostly insignificant; the branching factor of end-game positions is small, meaning even an unguided MCTS agent could find the optimal move through brute-force search. Sacrificing important strategic knowledge for the ability to fit numerous trivial positions accurately directly leads to the inverse scaling of Elo seen in Fig. 5.

## 8.3 Why do large models overfit?

The degradation of large model performance on early-game states is an example where RL scaling diverges from the quantization model. The quantization model explains how larger models could fit more tasks, without forgetting higher-frequency tasks. In RL, we see that fitting less-frequent states coincides with lower performance on the most frequent states, when states split into two distinct groups. The difference between supervised learning and RL scaling might be caused by a shift of the target distribution during training, which is known to cause training failure, e.g. through dormant neurons [44].

**Large models selectively forget.** During training, large models seem to forget what they learn about early-game states, but not about late-game. In Fig. 8 we plot the loss of the largest Oware model at different stages of training. We see a clear difference between the learning dynamics of early- and late-states: late-game loss improves gradually throughout training, while early-game loss improves, then degrades at late training.

A possible explanation of this selective forgetting is a selective shift in the distribution of the target labels, i.e. the value of each state. Sokar et al. (2023) [44] have shown that changes to the target distribution in RL can harm performance by creating dormant neurons. Unlike early-game states, late-game states have relatively fixed label distributions since the outcome is already decided by that point in the game. A hypothetical mechanism of inverse scaling then follows: 1) Models fit states by

order of frequency, gradually expanding the fraction of late-game states in the pool of learned states. 2) As new states are optimized, higher-frequency states learned earlier can be forgotten. 3) In Oware, late-game states are immune to forgetting due to their strongly fixed labels. 4) Large Oware models are trapped in local optima, fitting many end-game states well but forgetting a smaller fraction of crucial early-game states. See appendix H for a more in-depth explanation.

**Comparison to inverse scaling in supervised learning**   We note a certain similarity between this inverse scaling phenomenon and one of the causes for inverse scaling identified by McKenzie et al. (2023) [43], namely distractor tasks. Some LLM benchmarks require mastering a set of tasks $T$, but as model scale increases, models first learn a set of easier tasks $D$ that harm performance, leading to inverse scaling. Another similar concept is competing tasks, which can cause inverse scaling in a supervised learning setting [45]. While the concept of competing/distractor tasks bears resemblance to the Oware case, the mechanism behind these phenomena is different. These examples of inverse scaling in supervised learning are driven by the similarity of tasks, while in our case early- and late-game states are far from similar, which is why we believe inverse scaling is caused by indirect interaction between tasks through forgetting.

**Limitations.**   We note two limitations of our analysis of inverse scaling. First, unlike Oware, the influence of late game configurations is harder to visualize in Checkers. Since the transition to late-game states is smoother in Checkers (Fig. 6**B**), the Zipf curve only visibly skews at very high rank numbers (Fig. 5**C**). As a result, one would need exceedingly large amounts of data to accurately plot the rank at which late-states dominate the distribution.

The second limitation is the use of game states as proxies for independent task quanta. Oware inverse scaling cannot be solved simply by removing all late turns from the training distribution, possibly because the group of tasks that models tend to overfit only partially correlates with late-game states. Fig. 7 misses some of the dataset due to states that only appear once in training. These states are at the tail-end of the frequency curve, given their low individual frequencies, but they constitute the majority of states encountered in training. In appendix I we visualize data frequency over a different axis and show that the tail-end of the state distribution, which we removed for our above analysis, is crucial for accurately calculating frequency scaling along other axes. A better grouping of task quanta might allow one to know which states to exclude from training in order to prevent inverse scaling.

# 9   Discussion

We presented evidence connecting the quantization model of LLM neural scaling to AlphaZero power laws. In particular, we found that AlphaZero data follows Zipf's law and that agents optimize states according to their frequency rank, counter-intuitively achieving worse loss on easier states. We also shed light on the mechanism behind inverse scaling, showing that inverse scaling correlates with an abnormal frequency distribution. This in turn can cause agents with larger capacity to focus on end-game states, getting better loss on these states but worse loss on crucial opening moves.

While this work touches several aspects of scaling law theory, we still lack a full model of RL scaling that can fully connect state distribution to Elo scaling laws. We provided clear evidence of the effect of Zipf's law on loss, but loss is only a proxy for performance and the multi-agent Elo metric is connected to it in a non-trivial way. A significant problem with applying the quantization model to explain scaling laws is the difficulty of identifying task quanta, both for LLMs and RL, which is why we use game states as a proxy of tasks. Identifying the tasks or concepts that AlphaZero learns is a challenging interpretability problem, mostly limited to human game knowledge [31, 46].

We hope the analysis presented here will help advance the understanding of RL scaling. RL methods are often challenging to scale, and understanding why is a prominent topic in the field today [47, 48, 44, 49]. Our results hint at a possibility for improved RL algorithms using a curriculum informed by the frequency distribution, similar to recent attempts in supervised learning [33].

# Acknowledgments and Disclosure of Funding

We thank Eric J. Michaud and David J. Wu for insightful feedback and discussions.

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

# A   Additional Background

**AlphaZero**   AlphaZero is an RL algorithm for learning two-player zero-sum games, learning solely from self play [20]. The algorithm performs a Monte-Carlo tree search on the tree of future moves, guided by a neural network. The neural net receives the current board state as input, providing an estimate of its value, $v$, together with a policy prior, $\mathbf{p}$. The loss function is composed of separate elements for $v$ and $\mathbf{p}$:

$$L = (z - v)^2 - \boldsymbol{\pi}^\top \log \mathbf{p}. \tag{6}$$

During training, the agent visits board states $s$, having ground-truth labels $z$, for which it generates estimates $v$ and $\boldsymbol{\pi}$ for the state value and target prior, respectively. The MSE value-loss term $(z - v)^2$ pushes $v$ to fit the observed game outcome $z \in \{1, 0, -1\}$, representing win, draw and loss respectively. The cross-entropy policy loss $-\boldsymbol{\pi}^\top \log \mathbf{p}$ pushes $\mathbf{p}$ to fit the actual policy $\boldsymbol{\pi}$ generated by the tree search.

**AlphaZero scaling laws**   AlphaZero scaling is one of the earliest examples of neural power-law scaling in RL [21, 5]. Jones [21] showed that AlphaZero agents trained to play Hex improve their Elo score $r$ with the log of training compute $C$, $r \propto \log(C)$, with a coefficient that remains constant across board sizes. As they point out, this predictable scaling behavior allows one to plan the compute budget ahead of training to fit a desired performance level. Neumann & Gros [5] found similar compute scaling in agents trained on Connect Four and Pentago, as well as scaling with neural network size, and pointed out that these scaling curves are power laws of the Bradley-Terry playing strength $\gamma$ [35], since $r \propto \log_{10}(\gamma)$. Similar to LLM scaling laws, these scaling laws of AlphaZero allow one to know in advance the optimal model size that would achieve maximal Elo score, given the available training compute budget. An interesting result from Jones shows that training compute can also be traded off predictably for inference compute [21].

# B   Board game descriptions

We describe here the rules of all four games examined in this paper. All games are two-player, zero-sum, open information games, where players alternate turns between them moving pieces on a board.

**Connect Four**   The players alternate placing a disk of their color into one of seven vertical columns on a 6x7 grid, and the disk falls to the lowest unoccupied position in that column. A player wins by arranging four of their disks in a row, either vertically, horizontally, or diagonally. If all cells are filled without such an alignment, the game is drawn.

**Pentago**   Players alternate placing one marble of their color on any empty cell of a 6x6 board, which is divided into four 3x3 rotating quadrants. After placing, the player must rotate one quadrant by 90ř either clockwise or counter-clockwise. A player wins immediately upon obtaining five marbles in a row (horizontal, vertical, or diagonal) either before or after rotation. If the board becomes full without five in a row, the game is drawn.

**Oware**   This game is played on a board with two rows of six houses and optionally store pits. Each of the twelve small houses starts with four seeds. Each player controls the six houses on their side. On their turn, a player selects one of their houses with seeds, removes all seeds, and sows them counter-clockwise, placing one seed per subsequent house (excluding stores and skipping the original house when it's revisited). If the last seed lands in an opponent's house and brings its count to exactly two or three, those seeds are captured into the player's store. Capture continues backward from that house as long as the immediately preceding opponent's houses also contain exactly two or three seeds. Capturing all of an opponent's seeds is forbidden; in that case no capture is made. The game ends when a player captures 25 or more seeds, or both players capture 24 each (draw), or if no legal move exists; remaining seeds are then collected by their owners. In our analyses, a draw is also declared if the game lasts 1000 turns.

**Checkers**   The players each control 12 pieces on opposite dark squares of an 8x8 chess board. Normal pieces move diagonally forward one square to an unoccupied dark square. If an opponent's

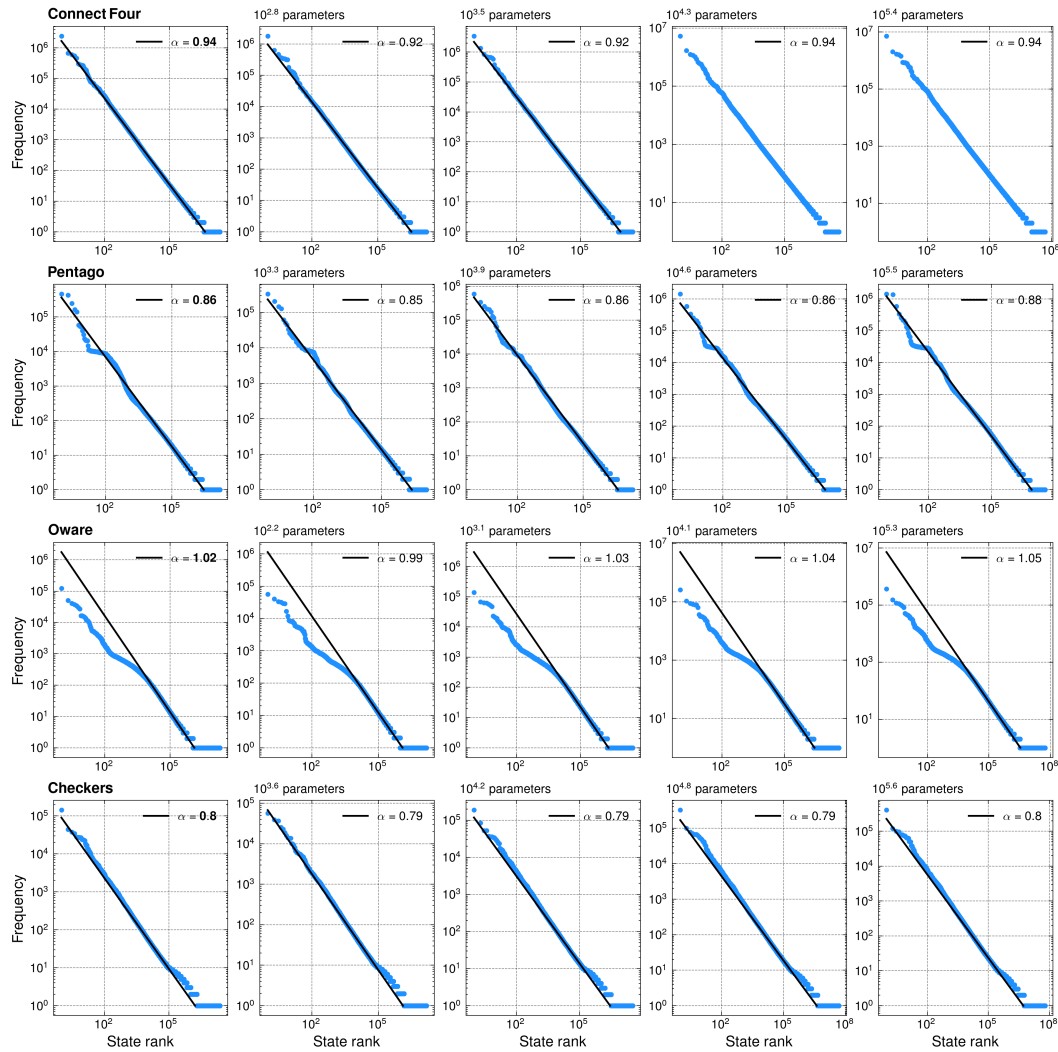

Figure 9: **Zipf's law curves for AlphaZero games.** On the left column, we plot the frequency distribution of states collected from different-sized agents training on each of the four board games discussed in this paper. In each row we plot curves describing the distribution of games played by single agents. We find small variations of the power-law exponent between agents, overall staying close to the general exponent of each game. The averaged exponents vary in a range that is in agreement with human-game exponents.

piece is diagonally adjacent and the following square is empty, the adjacent piece may be jumped and removed; multiple jumps are allowed in sequence if available. When a piece reaches the farthest row on the opponent's side, it is crowned as a king and gains the ability to move and jump both forward and backward. Play continues until one player has no legal move (draw) or a player lost all their pieces. We also draw the game if it lasts too long.

## C   Zipf's law variation

### C.1   Zipf-exponent variation among agents

The different strategies learned by individual agents lead to small differences in their frequency-distribution curves. Here we show examples of Zipf curves obtained from games played by individual agents during training. Agents follow slightly different Zipf laws, but the deviation between agents is smaller than the deviation between environments.

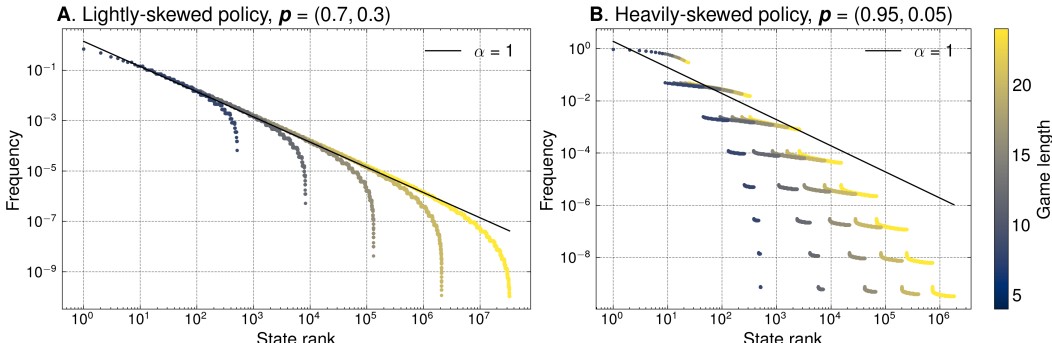

Figure 10: **Zipf's law in random games with a non-uniform policy. A:** In a toy-model game with a branching factor of 2, a policy that prefers one action over the other will smooth-out the plateaus of Fig. 2**A** into a Zipf's law with exponent $\alpha = 1$. The power law dies out near the end of the distribution, which depends on game length. **B:** If the policy heavily prefers one action over the other, it produces a series of plateaus again. These plateaus quickly diverge from the $\alpha = 1$ Zipf's law in short games, but converge to a Zipf's law as game length increases.

**Total Zipf distribution**   In Fig. 9 we plot the state-frequency distributions of different games, plotting the combined distribution of many agents in the left-most column. We do this by counting the total frequency of states across games played by several different-sized agents. Each environment has its own Zipf exponent, but all exponents lie in the range $[0.75, 1.05]$. For Oware, we fit the part of the curve that comes after the bump caused by late-game states, see appendix G. Checkers also exhibits a bump in the distribution, albeit harder to see since it starts at a much higher rank.

**Single-agent Zipf distribution**   Another set of Zipf curves can be obtained by counting state frequencies only in games played by a single agent. In each column after the left-most one, we plot the distribution generated by an agent with a different number of neural-net parameters. The standard deviation of the exponent among agents is low in all environments, standing at $0.01$ or lower for Connect Four, Pentago and Checkers, and at $0.02$ for Oware. The uniformity of the exponent across different agents agrees with prior observations on human games: the opening frequencies of human Go games also follow Zipf's law, with the same exponent fitting both amateur games and games played at prestigious Go tournaments [30].

## C.2   Zipf's law in non-uniformly-random games

Zipf's law in games arises from their branching-tree structure. In section 5 we discuss the origin of Zipf's law, showing that random toy-model games create a frequency distribution of plateaus centered on a power-law with exponent $\alpha = 1$. We show that this distribution can be smoothed into a Zipf's law by adding complicated game rules, but keeping the games random.

Here we show another mechanism that can smooth the plateau distribution into a Zipf's law. If the policy generating the games is not uniformly random, it generates smooth power laws. However, if the policy approaches a deterministic policy, the frequency plateaus reappear.

**Policy bias smooths the distribution**   In Fig. 10 we plot the state-frequency distribution of toy-model games under different agent policies. These games have a constant branching factor of 2, and last a fixed number of turns. We look at games generated by random policies, that are skewed towards preferring one action over the other at every level of the game tree.

Shifting away from the uniform policy $\boldsymbol{p} = (0.5, 0.5)$, we see that the plateaus of the uniform case become shorter and denser, eventually becoming a smooth curve. The distribution follows the $\alpha = 1$ power law, diverging only at the very end of the distribution. The power law stretches over more orders of magnitude as game length is increased.

**Extreme bias reintroduces plateaus**   Surprisingly, changing the policy too far from uniform eventually leads to frequency plateaus again. In Fig. 10**B** we plot games played by a policy that is

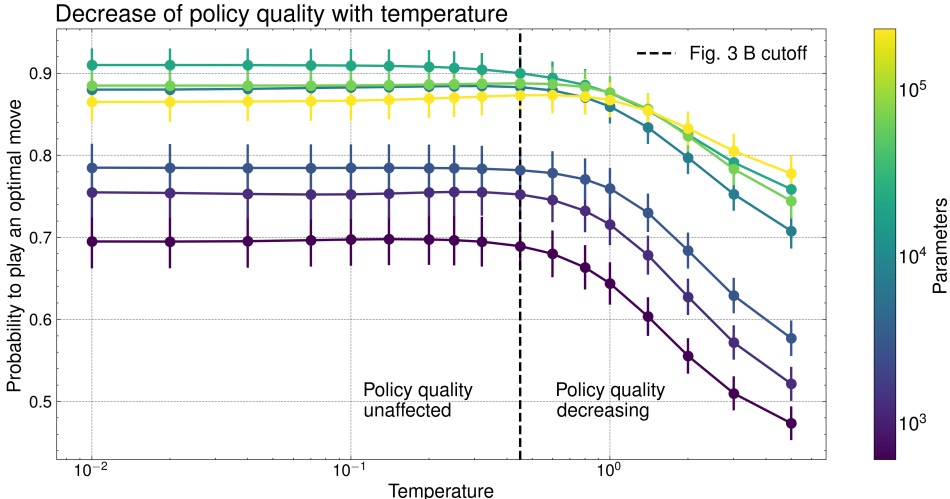

Figure 11: **The effect of temperature on agent performance.** We plot the probability that an agent will play an optimal move, against policy temperature $T$. As $T$ is increased, the probability to take the best action remains constant, starting to drop only above $T \approx 0.5$. The region where performance is unaffected is the same region plotted in Fig. 3**B**, confirming that the relation between Zipf- and scaling-law-exponents is causal, rather than the result of degrading performance due to high $T$.

close to being deterministic. The strong preference of one action creates plateaus of common games: the first plateau corresponds to an entire game where the same action was played every turn, the second plateau is made of games where the less-preferred action was played once, and so on. The plateau distribution still follows the linear $\alpha = 1$ trend, but diverges from it at a much earlier point than more-uniform policies. One must increase game length significantly to see that this distribution converges to a linear trend at the infinite-game-length limit.

## D   Policy degradation with temperature

In section 6 we plot the correlation between the scaling-law and Zipf's-law exponents by modulating the temperature of the action-selection policy, defined in Eq. 5. We claim that this correlation, plotted in Fig. 3**B**, is caused solely by the shifting Zipf distribution that affects agent performance, and not by temperature causing agents to play suboptimal moves.

Here we back up this claim by demonstrating that agent performance on individual states does not change in the temperature range used to plot Fig. 3**B**. To estimate performance on states, we look at the policy $\boldsymbol{\pi}(T)$ produced by Monte-Carlo tree search and plot the probability it assigns to optimal actions. For each state $s$, we calculate the probability the agent plays an optimal action:

$$\boldsymbol{p}(optimal) = \sum_{a \in A(s)} \boldsymbol{\pi}_a(T) \tag{7}$$

The set of optimal actions $A(s)$ is calculated with a Connect Four solver using alpha-beta pruning. An optimal action is defined as any action that, under optimal play, will lead to a victory, or to a draw if a victory is not possible. States where every move leads to a loss do not have any optimal actions, and are ignored in this analysis. The probability to "blunder", i.e. lose the lead in a game, is therefore equivalent to $1 - \boldsymbol{p}(optimal)$.

**Temperature does not harm performance in the examined range**   In Fig. 11 we plot $\boldsymbol{p}(optimal)$ against policy temperature $T$ for several Connect Four agent with different sizes[5]. For temperatures below the vertical dashed line, marking the region where exponents were calculated for Fig. 3**B**, the

---

[5]Error bars are the standard error of the mean.

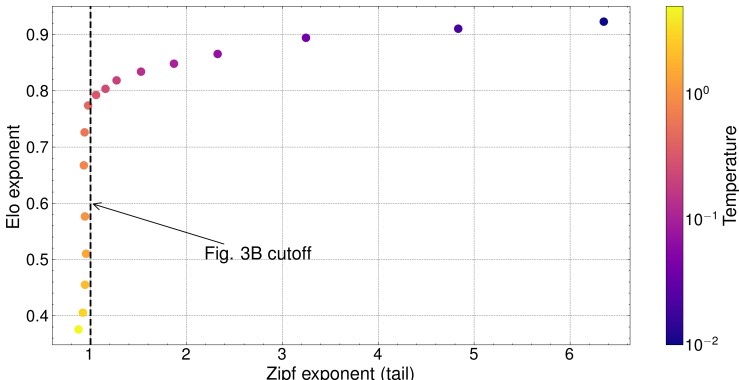

Figure 12: A zoomed-out version of Fig. 3**C** that includes higher temperatures (= smaller exponents). Fig. 3**C** only contains data to the right of the vertical cutoff line, which marks the temperature where $T$ stops affecting policy quality (equivalent to the line in Fig. 11). This line happens to coincide with the point where the Zipf exponent starts changing.

probability to take an optimal action is constant and does not change with $T$. This confirms that the change of the scaling-law exponent in this region is only due to the effect $T$ has on the Zipf curve.

It is also possible to notice the effect of temperature on the scaling laws in Fig. 3**B**. The scaling law exponent changes rapidly for $T > 0.5$, but slows down near $T \approx 0.5$. It then starts to change more rapidly only when the Zipf exponent starts to change significantly.

**Zipf's law only changes when policy quality is constant**   The full effect of temperature on exponent correlation is seen in Fig. 12, where we plot a zoomed-out version of Fig. 3**C**. The dashed line marks the border of Fig. 3**C**, which is the same point where temperature stops affecting policy quality in Fig. 11.

Interestingly, the vertical line in Fig. 12 marking the $T$ value where policy stops affecting policy quality coincides with the point where the Zipf curve starts to bend, as can be seen when looking at the change of the x-axis values. Policy quality changes significantly above $T \approx 0.5$, but the Zipf distribution changes significantly only below $T \approx 0.5$.

## E   Ground-truth loss scaling on different distributions

We present here another plot of ground-truth value-loss scaling, in addition to Fig. 4**B**. In Fig. 4**B**, we plot the loss of agents on a collective dataset, aggregated from the training runs of several agents of varying sizes. Here in Fig. 13**A** we plot the same loss, this time by calculating each agent's loss on their own dataset. The only difference between the two plots is the order of states; the ground-truth labels of each state remain the same. The error bars of Fig. 4 and Fig. 13 are the standard-deviation of agent loss across agents, either geometric when the y-axis is in logscale (Fig. 4**A**) or arithmetic when the y-axis is linear.

## F   The difference between AlphaZero value and actor-critic value

In sections 7 and 8 we present results on AlphaZero loss scaling, focusing on the loss of the network's value head. We chose the value head since it directly dictates the agent's policy both at training and inference time. This appendix, aimed for readers unfamiliar with AlphaZero, clarifies how AlphaZero's value head differs from the value net of actor-critic methods, which has no effect on policy at inference time.

Actor-critic methods are a family of popular policy-gradient algorithms, such as PPO and SAC [50, 51, 52]. These models use an architecture where two neural network outputs are used, namely a value estimation $v$ (or alternatively a $Q$ value) and a policy vector $\boldsymbol{p}$. Actions are sampled from the

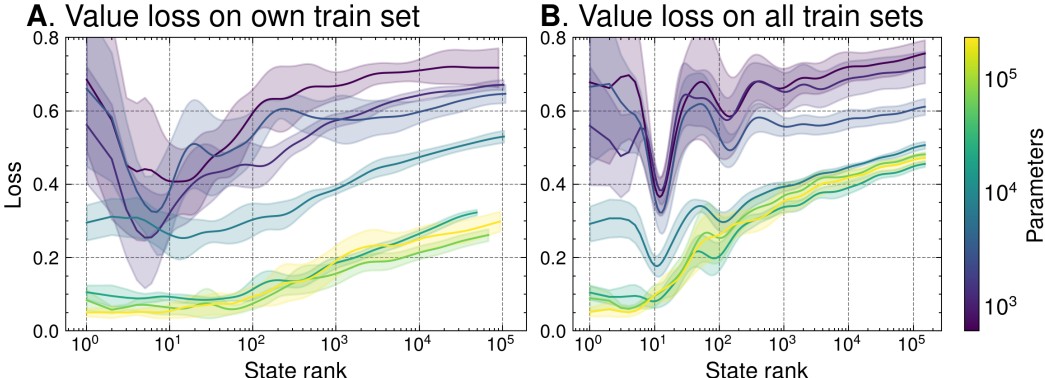

Figure 13: **Connect Four ground-truth value-loss on different datasets. A:** Average loss on each agent's own training data. **B:** Fig. 4**B**, presented here again for comparison, displaying average loss on a common dataset collected by different agents.

policy vector, which in turn is trained on the output of the value net $v$. While the loss of the value net on test data should be low for better agents, it has no effect on the actions played at test time.

In contrast, AlphaZero's value head is the main parameter used for choosing actions, while the policy head plays a secondary role as a static prior for Monte-Carlo tree search (MCTS). The AlphaZero policy $\pi$ is *not* the output of the policy head. Rather, $\pi$ is calculated using Eq. 5, repeated here for clarity:

$$\pi(s_0, a) = \frac{N(s_0, a)^{1/T}}{\sum_b N(s_0, b)^{1/T}}. \tag{8}$$

The visit count $N(s, a)$ is the number of times action $a$ was probed during MCTS, sampled in this equation for each action at the root node, meaning at the current game state $s_0$. The visit count is indirectly determined by the neural net's value and policy outputs $(v, \boldsymbol{p})$, since these parameters are used by MCTS to decide which future actions to probe when performing rollouts. The decision what action to probe at each junction in the game tree is done by maximizing the following quantity [53]:

$$\text{MCTS action} = \underset{a}{\text{argmax}}\left(Q(s, a) + U(s, a)\right) \tag{9}$$

The exploitation term $Q(s, a)$ is the average value $v(s')$ of all states $s'$ stemming from the state-action pair $(s, a)$. It is approximated by averaging the value-head output over all daughter nodes probed by MCTS so far:

$$Q(s, a) = \frac{\sum\limits_{\text{daughter nodes } s'} v(s')}{N(s, a)}. \tag{10}$$

The exploration term $U(s, a)$ is a variation of the PUCT algorithm [54], favoring actions with low visit counts $N$:

$$U(s, a) = c_{puct} \cdot \boldsymbol{p}(s, a)\frac{\sqrt{\sum_b N(s, b)}}{1 + N(s, a)}. \tag{11}$$

This term is weighted by the policy-head output $\boldsymbol{p}$, which serves as a constant prior that favors exploring some actions over others.

Low value head loss is a strong predictor of a good policy $\pi$, since it means the value head produces an accurate exploitation term $Q(s, a)$. The most frequently probed actions will be those with a high

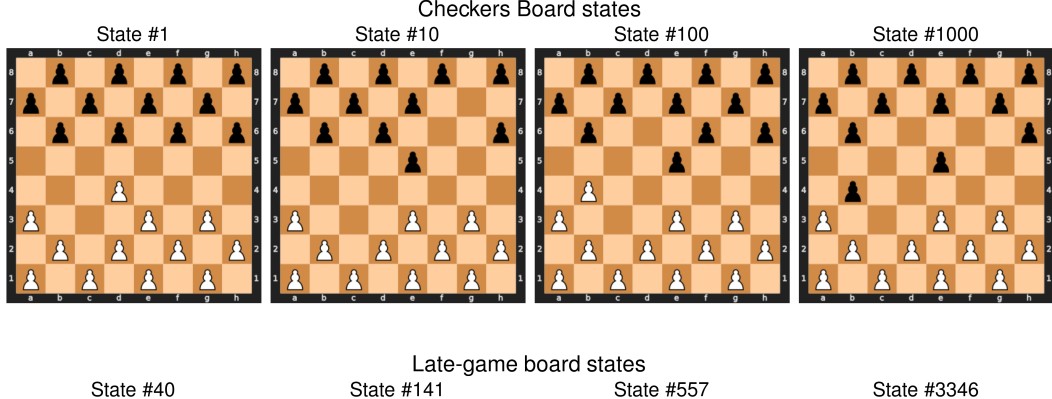

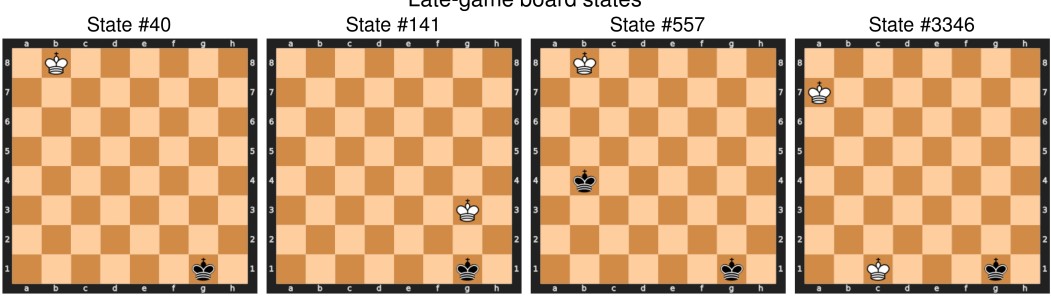

Figure 14: **Checkers board positions.** Low-rank (= high-frequency) states are mostly opening moves, visited shortly after the game begins. These states are mixed with high-frequency end-game positions, where only about 2-3 pieces still remain on the board, all promoted from man to king. The time it takes to end the game after reaching such an empty-board state is long; about half of all games end in a draw after no capture occurred for 40 turns.

$Q(s, a)$ value. In contrast, the policy prior controls an exploration term that is only significant at early stages of search, and vanishes at the limit of large search:

$$U(s, a) \xrightarrow[N \to \infty]{} 0 \qquad (12)$$

**Intuitive explanation**   We illustrate the difference between AlphaZero and actor-critic algorithms with a simple example. A trained actor-critic agent will be unaffected if we replace its value net with a random function at test time. In contrast, replacing the value head of AlphaZero with a random function at test time will generate an almost random policy; replacing the policy head with a uniform distribution will only somewhat degrade the policy.

## G   Board position visualization

We provide here visual examples of the two groups of high-frequency states found in Oware and Checkers. As we show in Fig. 6, late-game states get mixed with early-game states in the set of high-frequency states. This happens because by late game most pieces are captured, and the number of possible board states narrows down to a small selection.

In Figs. 14 and 15 we plot randomly-sampled game states from the state-frequency distribution, for both Checkers and Oware games. High-frequency states split into two clusters, namely opening moves and end-game moves. Early-game openings, plotted in the top row of each figure, are all small deviations from the initial board state. In the initial state of Checkers, each players' man pieces occupy their 3 closest rows; in the Oware initial state, there are 4 seeds in each pit.

The bottom row of each plot shows examples of late-game states. In these states, only a few pieces are still left on the board, after most pieces have been captured by the players. In Checkers, these states contain only 2-3 pieces that have already been promoted to king; in Oware, only a few seeds remain uncaptured. It is easy to see why these states have such high frequencies: most games end in one of those configurations, and the number of such possible configurations is very small. For example,

Figure 15: **Oware board positions.** States below rank 100 are all openings, deviating only slightly from the initial position of 4 seeds in each pit. At rank $164$, end-game states start to dominate the distribution, making up $> 80\%$ of all states. Most seeds have been captured in these late-game states, and the game ends shortly after they are played, when the active player has no seeds left on their row.

the number of possible Checkers configurations with two kings is 992, and the number of Oware configurations with three seeds is 1728 (ignoring player scores). Moreover, these configurations are not played with equal probability, leading to some states having even higher frequency.

## H   Inverse scaling mechanism

In section 8.3, we mention a hypothetical mechanism that could cause the observed inverse scaling phenomenon. Here we discuss this mechanism in more detail.

We make the following main assumptions:

- Agents fit states in descending order of frequency. An agent with a neural network of size $N$ can perfectly fit the values of $n = N/c$ states, where $c$ is the neural capacity needed to fit a single state [12].
- As the agent improves during training, game outcomes (i.e. state values) change. The values of early-game states change significantly during training, while the values of late-game states change only slightly, if at all.
- The shift of state values causes neural plasticity loss due to dormant neurons [44].

The first assumption is based on the quantization model of neural scaling [12].

The second assumption is based on the fact that, in most board games, player skill greatly influences the value of early-game states. In the games discussed in this paper, a match between random agents will usually end in a draw ($v = 0$) while a match between optimal agents will end in a first-player victory ($v = \pm 1$), except for checkers. In contrast, late game states that are played a few steps before the game ends have values that are largely independent of player skill, due to power imbalance. Very late states have fixed values if MCTS can probe the entire game tree stemming from them.

The third assumption is based on the work of Sokar et al. (2023) [44], who showed that changing the labels of a dataset during training causes some neuron activations to die out. These so-called "dormant neurons" effectively reduce the neural capacity of the model.

Assume we train a small agent with $10^m$ neurons and a large agent with $10^k$ neurons, $k > m$. In our model, the small agent roughly fits $10^m/c \approx n_{early}$ states, where $n_{early}$ are the number of early-game states with higher frequency than late game states. In Oware and Checkers, $n_{early}$ falls in the range $[100, 200]$, see Fig. 6. The large agent fits $10^k/c \gg n_{early}$ and mostly memorizes the values of late-game states.

For simplicity, let us treat the change of state values as a single, abrupt event during training. After this event, both models have high loss on early-game states, but maintain the same loss as before on late-game states, since the latter group kept their value labels. Due to plasticity loss, both models can now fit fewer states than before. The small agent now has high loss, since it could only fit the first $n_{early}$ states before values changed, meaning most of the states it memorized now have new labels. The large model, on the other hand, maintains a lower loss, since it could fit orders of magnitude more states than the first $n_{early}$ states, most of which are late-game states.

It is likely that the small agent, which now has loss close to that of a random agent, will continue to train and re-learn as many states as it can, starting from the most frequent ones. Eventually it will regain low loss on most or all of the first $n_{early}$ states. In contrast, the large agent maintains low loss, and could be close to a local minimum of the loss landscape that fits late-game states well but assigns wrong labels to early-game states. To leave this minimum and re-learn the first $n_{early}$ states, the large agent will have to sacrifice knowledge of some late-game states, since it lost a fraction of its neural capacity. In this scenario, the large model could remain stuck in the local loss minimum and never regain performance on early-game states.

In reality, it is more likely that value labels shift gradually rather than abruptly, meaning that loss increases gradually during training rather than spiking abruptly. We observe such a loss degradation in Fig. 8, where large agent value loss on early-game states starts to degrade mid-training.

# I   Frequency scaling on a different axis: capture difference

The quantization model of neural scaling laws states that models will learn independent task quanta by descending order of frequency. In language modalities, it is assumed that the known Zipf's law of word frequencies will lead to a Zipf's law of tasks. Similarly, we show that board states follow Zipf's law, suggesting that AlphaZero task quanta will also scale with a frequency power law. The states themselves are not independent quanta, since many of them share similarities, and strategic knowledge of one state can often transfer to another.

Here we show a different way to visualize training data frequency, to illustrate that more than one dimension of the data can have clear frequency scaling. When models fit tasks by frequency, some most-frequent tasks will not be represented at all on the state-rank curve, because states do not correspond to tasks directly.

To illustrate the high-dimensionality of the training data, we plot in Fig. 16 the frequency of capture-differences in Oware and Checkers board states. Capture difference is defined as the difference between the number of pieces captured by each player at a certain point in the game, in absolute value. We see that frequency drops smoothly in log-scale with capture difference, as states with a higher score-difference between players are rarer. According to the quantization model, one would expect agents to fit states according to the exponential distribution of capture-differences, giving exponentially-decreasing importance to states with higher differences.

The distribution of capture-differences cannot be represented correctly in our state-frequency analysis. We demonstrate this in Fig. 16, where we plot in yellow the same distribution, but omitting the tail of the Zipf's law state-frequency distribution. The tail is mostly composed of one-time states, i.e. states that appeared only once in training. In fact, the majority of unique states in the dataset have a frequency of 1. By discarding these states, as we do in our main results, we lose information about the exponential decay of capture-difference frequencies. Our main results cannot take these states into account, because exponentially-more games must be played in order to correctly measure the frequencies of one-time states.

Capture difference is an arbitrary measure, but it showcases the difficulty of visualizing data frequency. By visualizing one dimension of the data, we lose useful information about the frequency of other properties. If task quanta are indeed learned by the models, then these tasks would likely have a non-trivial representation, and visualizing their frequency could be difficult.

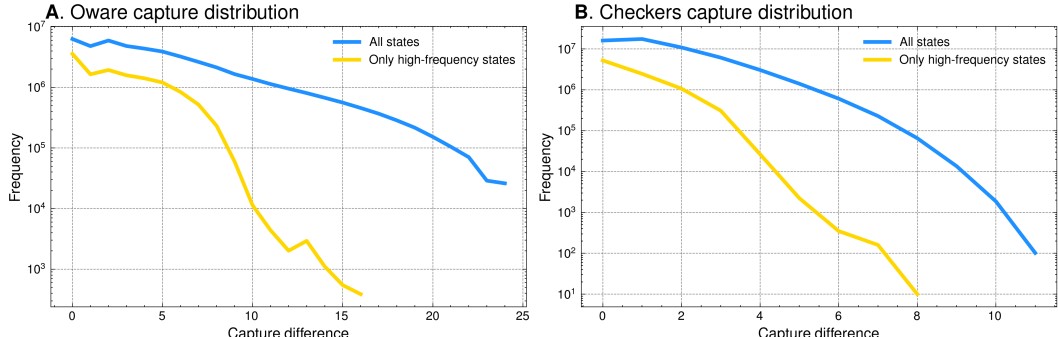

Figure 16: **State frequency distribution by capture difference.** Measuring the score difference between players provides another natural axis along which frequency drops smoothly. Agents could potentially prioritize optimizing small-difference states over large-difference, modelling more states with increasing capacity. *Prioritizing states in this way is **not** equivalent to prioritizing states by their frequency-rank, i.e. by the Zipf's law of Fig. 1, since a large portion of the states come from the tail-end of the Zipf distribution **which is discarded** in the main section results.* If we ignore the tail of Fig. 1 by plotting only high-frequency states, the frequency distribution changes significantly (yellow). It is likely impossible to visualize the true order in which data is learned with a simple metric.

## J   Distribution of ideal games

In section 5 we claim that random games played in a toy-model setting will result in a frequency distribution shaped as a series of plateaus, centered around a power law of exponent $\alpha = 1$. We verify this claim here.

In short, in a simple branching game the number of possible states is exponential in the turn number $t$, and all states at turn $t$ have an equal probability to appear in a game, hence the series of frequency plateaus observed in Fig. 2**A**.

**Frequency distribution function**   Consider a random game setting following the rules presented in section 5, namely:

- Each game lasts $K$ turns.
- Each turn $t$, a move $a_t$ is sampled uniformly from $b$ options.
- Each board position $s$ played at turn $t(s)$ can only be reached by a single, unique sequence of moves $\boldsymbol{a}_s = \big(a_1, a_2, \ldots a_{t(s)}\big)$. This means that one can define a state $s$ only by the sequence of moves played to reach it.

The position at turn $t$ is effectively sampled uniformly from $b^t$ possible positions, since the $t$ moves $a_i, i \in \{1, \ldots, t\}$ that lead to it are each sampled uniformly from $b$ options. Since $K$ turns are played each game, when sampling a board position $S$ randomly from all games played the frequency of sampling board state $s$ is:

$$P(S = s) = P(t = t(s)) \cdot P(S = s | t = t(s)) = \frac{1}{K} \cdot \frac{1}{b^t}, \tag{13}$$

where $t(s)$ is the turn number of board state $s$ and $P(t = t(s))$ is the probability to sample from turn $t(s)$.

Sorting the board positions in descending order of frequency results in a series of plateaus, each corresponding to a turn number. The number of board states in plateau number $t$ is $b^t$. Plateau number $t$ starts after board position number $n_{start}(t)$, defined by:

$$n_{start}(t) = \sum_{i=1}^{t-1} b^i = \frac{b^t - b}{b - 1}, \tag{14}$$

since each preceding plateau $i$ contains $b^i$ states. Board state number $n$ belongs to plateau number $t$ for the integer $t \in \mathbb{N}$ that satisfies:

$$n_{start}(t) \leq n < n_{start}(t+1) \,, \tag{15}$$

or:

$$\frac{b^t - b}{b - 1} \leq n < \frac{b^{t+1} - b}{b - 1} \,. \tag{16}$$

Shifting terms in the inequality and taking the log we get:

$$b^t - b \leq \qquad\qquad (b - 1)n < b^{t+1} - b \tag{17}$$

$$b^t \leq \qquad (b - 1)n + b < b^{t+1} \tag{18}$$

$$t \log(b) \leq \ \log\left[(b - 1)n + b\right] < (t + 1)\log(b) \tag{19}$$

$$t \leq \frac{\log\left[(b - 1)n + b\right]}{\log b} < t + 1 \tag{20}$$

since $t \in \mathbb{N}$, this is equivalent to using the floor operator:

$$t(n) = \left\lfloor \frac{\log\left[(b - 1)n + b\right]}{\log b} \right\rfloor \,. \tag{21}$$

Using Eq. 13 one sees that the frequency of the $n$-th most common board state, $s_n$, is proportional to:

$$P(S = s_n) = \frac{1}{K} \cdot \frac{1}{b^{t(n)}} \,, \tag{22}$$

where $t(n)$ is defined in Eq. 21. This is the series of plateaus seen in Fig. 2**A**.

**Distribution is bounded around power law**   Let us define $\tilde{t}(n) \in \mathbb{R}$ as $t(n)$ without the floor operator:

$$\tilde{t}(n) = \frac{\log\left[(b - 1)n + b\right]}{\log b} \,. \tag{23}$$

Since $x - 1 < \lfloor x \rfloor \leq x$, $P(S = s_n)$ is bounded by:

$$\frac{1}{K} \frac{1}{b^{\tilde{t}(n)}} \leq P(S = s_n) < \frac{1}{K} \frac{1}{b^{\tilde{t}(n)-1}} \,. \tag{24}$$

Expanding the bounding functions we get:

$$\frac{1}{b^{\tilde{t}(n)}} = \frac{1}{b^{\frac{\log[(b-1)n+b]}{\log b}}} = \frac{1}{e^{\log[(b-1)n+b]}} = \frac{1}{(b - 1)n + b} \,, \tag{25}$$

leaving us with:

$$\frac{1}{K} \frac{1}{(b - 1)n + b} \leq P(S = s_n) < \frac{1}{K} \frac{b}{(b - 1)n + b} \,. \tag{26}$$

We therefore see that the frequency distribution $P(S = s_n)$ is bounded by two straight lines (power laws with power 1) with different coefficients, which appear as two parallel lines in log-log scale, as we observe in Fig. 2**A**.

