# OpenReview forum: "AlphaZero Neural Scaling and Zipf's Law: a Tale of Board Games and Power Laws"
_NeurIPS.cc/2025/Conference — NeurIPS 2025 spotlight_

### Official Review · Reviewer_Vr29 · 2025-06-29

**Clarity:** 3
**Significance:** 3
**Originality:** 3
**Rating:** 4
**Confidence:** 3

**Summary:**

This paper presents an empirical study of neural scaling in AlphaZero across four board games, showing that the frequency of self-play states follows a Zipf power law and that model performance (Elo and value-head loss) scales predictably with network size in accordance with a quantization framework. By varying MCTS temperature, the authors demonstrate a direct link between the Zipf exponent of state distributions and the size-scaling exponent of performance. They further reveal an inverse scaling phenomenon in certain games and show that larger models learn rarer states earlier. Together, these findings extend scaling-law theory from supervised learning to reinforcement learning and suggest that distribution-aware training curricula could improve RL scalability.

**Questions:**

1. How robust are your Zipf distributions and scaling links to changes in MCTS rollout count or exploration parameters?
2. Could you discuss the key challenges you anticipate and how you might adapt your analysis to handle games with imperfect information?

**Ethical Concerns:**

["NO or VERY MINOR ethics concerns only"]

**Limitations:**

Yes

**Quality:**

3

**Strengths And Weaknesses:**

**Strengths**:

I really enjoyed how thorough the authors were—testing across very different board games, scaling up network sizes, and tweaking MCTS temperature to show that Zipf’s law in state frequencies isn’t just a curiosity but actually drives the familiar power‐law scaling (and even the surprising inverse scaling) we see in performance. I also found the diagnosis of overfitting to those “end‐game” bumps especially interesting. Besides, the paper is well-written and easy to follow.

**Weaknesses**:
1. My main concern is the insufficient discussion about the related work. The paper cites Jones et al.’s “Scaling scaling laws with board games” only in the bibliography and never contextualizes how its Zipf-based findings compare conceptually or empirically.
2. While the authors note that loss is only a proxy for true playing strength, there is no formal model or empirical ablation connecting loss-rank curves to the multi-agent Elo metric.

---

> ### Author Rebuttal · Authors · 2025-07-31
>
> Thank you for your review. We are glad you enjoyed how thorough the paper is, and found our results interesting. We address the points you brought up in order:
>
> - We would like to point out that we do cite Jones [1] in the main section, as well as the appendix. Our paper is a direct extension of Jones' work. That being said, Jones only discusses the scaling form of AlphaZero and not the origin of scaling. Neither do Neumann & Gros [2]. We therefore find both [1] and [2] to be worthy of discussion as background rather than related work. We agree with you that the paper would benefit from discussing this background more thoroughly, which is why we added a new part to appendix A that covers this. You can read the new addition at the end of this comment. Ideally we would put this in the "Background" section, but due to page limit constraints we refer the reader to the appendix, the same way we cover AlphaZero (Silver et al. [3]).
>
> - Indeed we do not present a model connecting loss and Elo, as we point out in the discussion. Elo is a multi-agent metric that is not easy to connect to single-agent attributes like loss, let alone to performance on individual states.
>
>
> ### Questions
>
> - This is an interesting question, considering MCTS scaling is one of the main finds of Jones (2021).
> We did not directly perform an ablation study on MCTS rollouts, but we did measure the effect of exploration on Zipf's law in appendix sections B.2 and C.
> In appendix C we change the temperature and see that it significantly changes both the Zipf distribution and the scaling law, see figure 10.
> In appendix B.2 we look at random games and show that the state frequency distribution shifts significantly between a uniform policy (exploration) and a skewed policy (exploitation).
>
> - It should be possible to perform a similar analysis on imperfect information games like Stratego [4], Diplomacy [5] or Hanabi [6], although an extensive analysis might be hard since each of these games is solved by a different algorithm. Other than focusing on AlphaZero, our paper is agnostic to perfect vs. imperfect information, except for the state definition. In an imperfect information setting it might be reasonable to count the frequency of the observable states rather than look at the full hidden state.
> If one still wants to calculate value loss on the hidden states, and not the observable states, then one would have to account for irreducible loss since the state is effectively stochastic from the agent's point of view.
>
>
> We hope we have addressed all your concerns, and encourage you to look at other new sections we added to the paper found in our replies to reviewers c7VB, a847  and hdGQ. If we did, we hope you would consider updating your score. Otherwise, we would be glad to answer any remaining questions.
>
> ### References
>
> [1] Jones "Scaling scaling laws with board games." (2021).
>
> [2] Neumann and Gros. "Scaling laws for a multi-agent reinforcement learning model."  (2022).
>
> [3] Silver et al. "A general reinforcement learning algorithm that masters chess, shogi, and Go through self-play." (2018).
>
> [4] Perolat et al. "Mastering the game of Stratego with model-free multiagent reinforcement learning." (2022).
>
> [5] Bakhtin et al. "Mastering the game of no-press diplomacy via human-regularized reinforcement learning and planning." (2022).
>
> [6] Cui et al. "Adversarial diversity in hanabi." (2023).
>
>
> # New addition to appendix A: AlphaZero scaling laws
>
> AlphaZero scaling is one of the earliest examples of neural power-law scaling in RL [20,5]. Jones [20] showed that AlphaZero agents trained to play Hex improve their Elo score $r$ with the log of training compute $C$, $r \propto log(C)$, with a coefficient that remains constant across board sizes. As they point out, this predictable scaling behavior allows one to plan the compute budget ahead of training to fit a desired performance level.
> Neumann \& Gros [5] found similar compute scaling in agents trained on Connect Four and Pentago, as well as scaling with neural network size, and pointed out that these scaling curves are power laws of the Bradley-Terry playing strength $\gamma$ [33], since $r \propto log_{10}(\gamma)$.
> Similar to LLM scaling laws, these scaling laws of AlphaZero allow one to know in advance the optimal model size that would achieve maximal Elo score, given the available training compute budget. An interesting result from Jones shows that training compute can also be traded off predictably for inference compute [5].

---

> > ### Comment · Reviewer_Vr29 · 2025-08-08
> >
> > Thank you for your thorough response, which addresses most of my concerns and clarifies the treatment of related work and background. I continue to think this is an insightful paper that merits acceptance, but I still recommend improving the organization—particularly by integrating key background and analysis highlights from the appendix into the main text, as tracing related content (e.g., for Q1) currently requires significant effort. For these reasons, I am maintaining my rating of 4.

---

> ### Author Response · Authors · 2025-08-08
>
> Dear reviewer, since today is the last day of author-reviewer discussion, we would kindly ask you to respond to our rebuttal. We would be glad to address any remaining issues you may have.

---

> ### Author Response · Authors · 2025-08-08
>
> Thank you for the quick reply. We are glad you believe the paper merits acceptance, now that most of your concerns are addressed.
>
> Since it seems like your only remaining concern is the positioning of sections, rather than their contents, we would be glad to improve the paper structure with your help.
>
> The camera ready version allows for an additional page. We can therefore offer the following changes:
>
> 1. Move appendix C (Temperature and correlation with scaling laws) to the main section, to address your question 1.
> 2. (Alternatively) Move the background on Jones (2021) and scaling laws, together with Silver et al. (2017), from appendix A to the main Background section.
> 3. In both cases 1 and 2, we will add a comment to the main paper containing our response to your question 1 regarding the effects of exploration on Zipf's law.
>
> We would have liked  to apply both 1 and 2, but even the extra page is not enough to accommodate both additions. We would greatly appreciate your opinion on these suggestions.

---

### Official Review · Reviewer_cnDZ · 2025-07-01

**Clarity:** 3
**Significance:** 3
**Originality:** 3
**Rating:** 5
**Confidence:** 4

**Summary:**

- Observes that AlphaZero agents produce power law distributions for game states
- Compares this result with the quantization hypothesis proposed by Michaud et al. 2023
- Provide evidence from Pentago and Connect Four supporting a prior theory by Blasius & Tönjes (2009)
- Shows that models exhibit lower loss on more frequent states
- Studies inverse scaling and provides evidence that it can be caused by end-game states that are disproportionately prevalent

**Questions:**

Please see "Strengths and Weaknesses"

**Ethical Concerns:**

["NO or VERY MINOR ethics concerns only"]

**Final Justification:**

I think I overanchored on some details in the paper that either I misunderstood or perhaps were expressed suboptimally. Based on my discussion with the authors, I think the paper is quite solid.

**Limitations:**

Yes

**Quality:**

3

**Strengths And Weaknesses:**

## Abstract

- “Neural scaling laws are observed in a range of domains, to date with no clear understanding of why they occur.” This seems objectively false. I would strongly urge the authors to strike this sentence or explain in more detail why they think the numerous theoretical works on neural scaling laws are inadequate. Also, some somewhat recent theoretical work on neural scaling laws goes uncited: https://arxiv.org/abs/2405.15074 and https://arxiv.org/abs/2502.17578; these papers might have more theoretical citations worth including.

- Related to the above point, perhaps the authors are trying to suggest that the origin of neural scaling laws in RL are much less well understood theoretically than in other toy models? If so, that seems like a more defensible claim.

- Suggestion: The first 3-4 sentences are a bit rough. It might be simpler to just state, “In this paper we examine power-law scaling in AlphaZero, a reinforcement learning algorithm, using a theory of
language-model scaling. Specifically, we leverage the so-called quanta scaling theory, which posits “

- I’m not sure whether the “quanta scaling theory” should be called a theory. Is it a hypothesis or a theory? I learn more towards the former but I’m open to being persuaded.

## Introduction

## Background

- Figure 1: Regarding the claim “Board-state frequency follows a power law…”, one detail I’m confused by is that surely this is policy-dependent? Like, if my Connect Four policy is to always play in the left-most available column, one would find a different state-frequency distribution. Is this claim indeed policy dependent? If not, why not? If yes, then under what conditions does the policy yield a power law board-state frequency?

- Line 70: I know that Michaud’s paper is not your paper, but the condition “ If the loss on each task is reduced by a fixed amount ∆L after it is learned” seems like a strong condition. Similarly, “It is assumed that a fixed neural capacity c is needed to fit each of the independent task quanta” seems like a strong assumption again.

## Methods

- Line 121 “we present similar size-scaling curves for Oware and Checkers in Fig. 4A”. As best as I can tell, Figure 4A does not show power laws. Perhaps Fig 4B and 4C were intended?

- “unlike the chess analysis of Blasius & Tönjes (2009)”. Could the authors please clarify in what way Blasius & Tönjes assumed the history of previous moves?

## Zipf’s Law in AlphaZero

- Line 143 “This result strengthens the assumption that Zipf’s law is not caused by human decision-making, but is rather derived from the game rules.” This seems like an incorrect inference to draw. Maybe the power laws are attributable to AlphaZero as an algorithm. Or maybe the power laws are attributable to being trained for optimal play. At this point in the paper, I don’t think any evidence has been provided that justifies a statement like “Zipf’s law is caused by the game rules.”

- Line 142-147: After reading citation 27 (Blasius and Tonjes 2009), it becomes clear that (1) the cause of the power law scaling is clear, and that (2) its cause is not attributable to humans but to the structure of hierarchical fragmentation: “We propose a simple stochastic process that is able to capture the observed playing statistics and show that the Zipf law arises from the self-similar nature of the game tree of chess. Thus, in the case of hierarchical fragmentation the scaling is truly universal and independent of a particular generating mechanism.” Consequently, I feel that this paragraph does not accurately represent prior work and paints a misleading picture of a “puzzle” which has already been solved for over a decade.

- Lines 148 to 156 claim to offer *intuition* of the source of Zipf’s law, but I can find no intuition in either that paragraph or Figure 2. If the authors aim to offer intuition, please improve this exposition; otherwise, strike this language. Statements like “ is easy to show that an ideal game, with a constant branching
factor and a tree structure free of loops, would produce a Zipf-like distribution.” are not offering intuition; they’re just restating the claim.

- I’m not sure I understand how Figure 2A supports the claim of Zipf’s law. Rather, the plateaus suggest that the frequency distribution does not depend on the element rank n, but on whether n falls into some bucket. To maybe express this idea differently, when we say something follows Zipf’s law, how precisely do we mean?

- I’m not sure I understand Figure 2B. The power laws that are displayed for Connect Four and Pentago do not go through the plateaus, as shown in Figure 2A. Rather, the power laws appear to be an envelope over the plateaus.

## Analyzing how loss scales with Zipf’s Law

- Figure 3 C: “The time needed to evaluate the same states using alpha-beta pruning [37] drops exponentially”. I would not classify this as exponential. It looks pretty clearly linear in log-log space, at least from below 1e2 to 1e5. Same with line 190. I would not call Figure 3 C exponentially decreasing.

- Line 171 “we find that AlphaZero models reduce their loss on game states in descending order of frequency”. -> This is unusual phrasing. I think it is clear that models show lower loss on higher ranked states. But “reduce their loss” is ambiguously and possibly suggestive of learning dynamics. I think it’d be more clear to say “Models exhibit lower loss on more frequent states” or something like this.

- Line 174: “Comparison to LLMs” I think that this phrasing is misleading. The quantization hypothesis is a hypothesis that to my knowledge has never been confirmed/verified in language models. The paragraph is fine, but the leading phrase should say “Comparison to Quantization Hypothesis”

- Line 184 “It is surprising that absolute value loss (Fig. 3B) tends to increase with state rank,” -> I don’t find this surprising at all. Using an example of classification with imbalanced classes, we expect models to perform better on majority classes rather than minority classes because doing so minimizes the loss more; for an extreme example, if a class has exactly 0 samples, then there’s no reason for the model to learn about this class. Returning to this AlphaZero setting, the first few states appear in (nearly) every game, whereas the 1e5 state occurs much much much more rarely.

## Connecting Inverse Scaling to the Game Structure

- Line 264: Please stop calling the quantization hypothesis “LLM scaling theory”

---

> ### Author Rebuttal · Authors · 2025-07-31
>
> Thank you for your thorough review. We address the points you brought up in order:
>
>
> ### Abstract
>
> 1) (first two points) Thank you for the recent citations, we have added them into the text.
> Indeed, as you observed, our intention in the sentence “Neural scaling laws are observed in a range of domains, to date with no clear understanding of why they occur” was that no model of neural scaling can explain this scaling in all domains, including the RL domain. There are indeed good models for explaining power law scaling in supervised learning settings. To avoid confusion, we changed the wording of that sentence, from "no clear understanding" to "no universal understanding".
>
> 3) Thank you for the suggestion. Since the other reviewers did not comment on the abstract wording, we feel that the current phrasing is a good fit (including the change mentioned above). If you still feel strongly about the wording we are open to finding a good replacement.
>
> 4) We see the problem with the wording "theory". We now refer to it throughout the paper as the "quantization model", as is done in Michaud et al. [1]
>
> ### Background
>
> 1) We discuss this point thoroughly in section 5 ("Zipf’s law in AlphaZero"). It seems  that you have answered your questions yourself in your later comments on section 5 and Blasius & Tönjes [2], but if something is still unclear we will gladly clarify it.
>
> 2) As you point out, Michaud et al. [1] is not our paper. We do agree with you that the assumptions they make are somewhat strong, since they discuss a toy model.
>
> ### Methods
>
> 1) It seems our use of the word "similar" in line 121 lead to this confusion. We meant it in the sense that we calculate size-scaling curves in the same manner they were calculated by Neumann & Gros [3]. We removed the word "similar" to avoid confusion.
>
> 2) Blasius & Tönjes [2] discuss the frequency of chess openings, where "opening" means a sequence, or history, of specific moves.
>
> ### Zipf’s Law in AlphaZero
>
> 1) It seems you answered this question in your next point. We state the justification of line 143 in the rest of the paragraph, which we feel is adequate, namely that seeing Zipf's law in AI games strengthens the assumption that Zipf's law is not unique to humans, but is rather a general phenomenon for (optimized) play.
>
> 2) We explicitly state in the next paragraph (line 149) that Blasius & Tönjes already suggested (without proof) the connection of game trees to Zipf's law. Nowhere in the paper do we claim that we are the first to make that observation.
>
> 3) We thank you and reviewer c7VB for pointing this out. We now added an appendix section verifying our statement on Zipf's law in ideal games ("it is easy to show..."). Since it doesn't fit in this comment, we urge you to look at our comment to reviewer c7VB where we posted the content of the new appendix.
>
> 4) We never claim figure 2A is a Zipf's law, indeed calling it that would be wrong. It is a series of plateaus around a straight line, which is a "power law" with power 1. We use the term "Zipf-like" since this distribution is centered around a power law.
>
> 5) Figure 2B shows the Frequency distributions of random games. Indeed, these are not clean Zipf's laws, and not plateaus, but rather a transition point between the two. The power law fit we added there fits the mid-section of the Connect-Four distribution, ignoring the noise at low ranks and the finite-size effects at high ranks. Notice the plateaus at high ranks are just an artifact of our limited calculation, and will disappear at the limit of infinite data (and never-ending games). In contrast, the plateaus of figure 2A are not artificial and will not disappear at this limit.
>
> ### Analyzing how loss scales with Zipf's Law
>
> 1) We agree with you on this point, it looks like a power law would be the best fit to this trend. We meant "drops exponentially" as in "drops on an exponential scale", we now changed the wording to avoid confusion.
>
> 2) Thank you for the suggestion, we agree it is better worded. We incorporated it into the text.
>
> 3) We also incorporated this naming suggestion.
>
> 4) You touch here exactly on the point we make in this section: our results _are_ surprising _despite_ the fact that they would be trivial for a simple classification model. In your example, the model learns static labels through supervised learning. In our paper, labels (=values) are not static. The hardest labels to model correctly through self play are the labels of the most frequent states, as we show in figure 4C.
> To use your example, the first few steps do appear in nearly every game, but at the start of training they all have wrong labels, i.e. they have a label of roughly 0 (draw) instead of the ground-truth label, which can be 1 or -1. That is because two inexperienced players will usually draw, compared to optimal players who will always end the game with a first player win (excluding checkers). Compare this to end-game states, where the label is almost guaranteed to be accurate since MCTS will probe all possible future moves.
> To summarize, “It is surprising that absolute value loss tends to increase with state rank” since the model has to improve, then unlearn and re-learn the labels of the most frequent (early-game) states, before absolute loss will decrease.
>
> ### Connecting Inverse Scaling to the Game Structure
>
> 1) We acknowledge this in our reply to the last point in "Abstract".
>
> We hope we have addressed all your concerns, and encourage you to look at other new sections we added to the paper. Other than the one found in our reply to reviewer c7VB following your question, we also posted new material in our replies to reviewers hdGQ, a847 and Vr29. If we did address all your concerns, we hope you would consider updating your score. Otherwise, we would be glad to answer any remaining questions.
>
>
> ### References:
>
> [1] Michaud et al. "The quantization model of neural scaling." (2023).
>
> [2] Blasius and Tönjes. "Zipf’s law in the popularity distribution of chess openings."
>
> [3] Neumann and Gros. "Scaling laws for a multi-agent reinforcement learning model."  (2022).

---

> ### Comment · Reviewer_cnDZ · 2025-08-05
> **Response to Author(s)**
>
> Thank you for writing your rebuttal. I think a lot of good ground was covered here. I also think I misunderstood/overemphasized some language in the original manuscript that I hope the authors will modify.
>
> I'm raising my score to a 5.

---

### Official Review · Reviewer_a847 · 2025-07-01

**Clarity:** 3
**Significance:** 3
**Originality:** 2
**Rating:** 5
**Confidence:** 4

**Summary:**

This paper applies the quantization model of language model scaling to analyze the scaling behavior of AlphaZero in board games. The authors first empirically show that AlphaZero-generated data follows Zipf’s law and offer intuitive explanations for how this distribution arises from the tree-like structure of board game state spaces. Building on this, they investigate the scaling properties of AlphaZero across the state spectrum and reveal a surprising pattern: the models appear to optimize state prediction loss in descending order of state frequency. The paper also observes instances of inverse scaling in certain games and provides empirical insights linking these behaviours to the unique tree structures of those games.

**Questions:**

I have a few questions on the empirical design and analyses that need further clarifications:
* Regarding the loss vs. state rank plots used throughout the paper — since the state spectrum can vary across training runs or over the course of training, the specific state corresponding to a given rank may also change. Could this variation confound the analyses when comparing different curves? In particular, in Figure 7, while we observe an increase in loss for "early turns" as training progresses, could this trend simply be due to the change of the underlying set of states being measured, rather than a true degradation in performance on those states?
* In the scaling law analyses, the value loss is used as the primary metric. Is there a specific reason for choosing value loss? How would the results differ if policy loss or Elo score were used instead?
* Regarding the inverse scaling, do you think some double decent phenomenon could be observed if we further increase model sizes?

**Ethical Concerns:**

["NO or VERY MINOR ethics concerns only"]

**Final Justification:**

Overall, I think the paper provides a meaningful contribution, and the author response clarifies my questions regarding experiment setups. I will keep my positive rating for an acceptance.

**Limitations:**

Yes.

**Quality:**

3

**Strengths And Weaknesses:**

Strengths
* The paper provides a very first empirical study on the scaling behaviours of AlphaZero, a problem that may become increasingly important as similar algorithms may be applied in the training of frontier models
* The paper offers many insightful empirical analyses that may shed light on how AlphaZero models learn and scale, such as how the Zipf’s distribution emerges and how it connects to inverse scaling.
* The empirical analyses in the paper are extensive and sufficiently support the major claims in the paper.

Weaknesses
* The paper lacks a dedicated section that provides readers with sufficient background on the AlphaZero algorithm and the various board game configurations used in the study.
* The empirical analyses in the paper are somewhat limited to board game environments and specific design choices. However, I think this is acceptable for an initial exploration aimed at understanding the scaling behavior of AlphaZero. I detail some questions on the specific empirical design in the followup section.

---

> ### Author Rebuttal · Authors · 2025-07-31
>
> Thank you for your review. We are glad that you see the relevance of this paper to frontier model RL training, and that you find our analyses extensive and sufficient to support our claims.
> We address the points you brought up in order:
>
> - We do have a short appendix section (appendix A) describing the basics of the AlphaZero algorithm, but we agree the paper does not provide an overview of the board games we test. Following your suggestion, we now added a new appendix section describing all four board games. We appended the new section to the end of this comment.
>
> - Thank you for acknowledging the scope of our experiments is sufficient for the paper's purpose. We answer your questions below.
>
> ### Questions
>
> - This is a great question. We answer regarding figure 7 and in general:
> 	- About figure 7, we specifically used the same state distribution for all training-time curves in order to avoid the confounding factor of a shifting rank distribution. We aggregated the state distribution over the entire training period (excluding early training), and then calculated the loss of these states for each agent checkpoint.
> 	- In general, the state distribution does change between training runs and during training, which poses a problem. To reduce the variance of state rank ordering, we aggregated states only after 30% of training time has passed, to reduce noise from early-training exploration. When possible, we aggregated over different random seeds as well.
>
> - We plot loss rather than Elo score since there is no straightforward way to assign an Elo score to a single state. Between value loss and policy loss, we decided that value loss is a more useful metric for two reasons:
> 	1) Policy loss does not necessarily tend to zero as policy improves. For example, two maximally different policies can both be optimal when one than one action is optimal. For example, if $a=1$ and $a=3$ are both optimal actions, both policy $p=(1,0,0)$ and policy $p=(0,0,1)$ will be optimal, as well as any combination of the two. When using an optimal policy as reference for calculating loss, higher loss doesn't mean that the policy is far from optimal play.
> 	2) The value has more influence on performance than the policy in AlphaZero. This is a bit counter-intuitive for people more familiar with actor-critic models. Appendix section D explains this point in detail.
>
> - That could be possible, although we haven't found any evidence to support it. We suggest in section 7 that AlphaZero inverse scaling might be caused by loss of neural capacity through dormant neurons; at the infinite size limit, the model should have enough capacity to re-learn the values of early-game states after the labels shift.
>
> We would also encourage you to check out other new additions to the paper, found in our replies to reviewers c7VB, hdGQ and Vr29.
>
> # New appendix section: Board game descriptions
>
> We describe here the rules of all four games examined in this paper. All games are two-player, zero-sum, open information games, where players alternate turns between them moving pieces on a board.
>
> ### Connect Four
>
> The players alternate placing a disk of their color into one of seven vertical columns on a 6x7 grid, and the disk falls to the lowest unoccupied position in that column. A player wins by arranging four of their disks in a row, either vertically, horizontally, or diagonally. If all cells are filled without such an alignment, the game is drawn.
>
> ### Pentago
>
> Players alternate placing one marble of their color on any empty cell of a 6x6 board, which is divided into four 3x3 rotating quadrants. After placing, the player must rotate one quadrant by 90° either clockwise or counter-clockwise. A player wins immediately upon obtaining five marbles in a row (horizontal, vertical, or diagonal) either before or after rotation. If the board becomes full without five in a row, the game is drawn.
>
> ### Oware
>
> This game is played on a board with two rows of six houses and optionally store pits. Each of the twelve small houses starts with four seeds. Each player controls the six houses on their side. On their turn, a player selects one of their houses with seeds, removes all seeds, and sows them counter-clockwise, placing one seed per subsequent house (excluding stores and skipping the original house when it's revisited). If the last seed lands in an opponent's house and brings its count to exactly two or three, those seeds are captured into the player's store. Capture continues backward from that house as long as the immediately preceding opponent's houses also contain exactly two or three seeds. Capturing all of an opponent's seeds is forbidden; in that case no capture is made. The game ends when a player captures 25 or more seeds, or both players capture 24 each (draw), or if no legal move exists; remaining seeds are then collected by their owners. In our analyses, a draw is also declared if the game lasts 1000 turns.
>
> ### Checkers
>
> The players each control 12 pieces on opposite dark squares of an 8x8 chess board.  Normal pieces move diagonally forward one square to an unoccupied dark square. If an opponent's piece is diagonally adjacent and the following square is empty, the adjacent piece may be jumped and removed; multiple jumps are allowed in sequence if available. When a piece reaches the farthest row on the opponent's side, it is crowned as a king and gains the ability to move and jump both forward and backward. Play continues until one player has no legal move (draw) or a player lost all their pieces.

---

### Official Review · Reviewer_hdGQ · 2025-07-02

**Clarity:** 3
**Significance:** 2
**Originality:** 2
**Rating:** 5
**Confidence:** 3

**Summary:**

This paper studies scaling laws in AlphaZero for board games like Checkers, Oware, etc.. The authors show that due to the tree-structure, the game states follow Zipf's law wherein certain positions appear much more frequently than others and that the model learns frequent states first. The work also explains "inverse scaling" phenomena where larger models perform worse in games like Oware because they focus too much on frequent but unimportant late-game positions.

**Questions:**

1. What exactly is the surprising or unexpected finding here? Since Michaud et al.'s theory is modality-agnostic and requires only multi-task setting with Zipfian data, isn't finding Zipf's law in board games somewhat expected? It would be nice to have a much clearer distinction between: (a) novel empirical discoveries, (b) expected theory confirmations, and (c) new theoretical insights.

2. I am also curious if there are any games or text-based environments where “task quanta” can be defined much more clearly.

3. Another curious question I had is related to the last line of Conclusion section:

> “Our results hint at a possibility for improved RL algorithms using a curriculum informed by the frequency distribution, similar to recent attempts in supervised learning [31].”

Could the preference of large models for late-game states be due to confounding factors other than frequency? For e.g., late-game states might have inherently lower variance in their true values (since fewer future moves remain) or more stable training targets (since outcomes are nearly determined), making them objectively easier optimization targets regardless of how often they appear? Have you controlled for such intrinsic difficulty differences to isolate the frequency effect?

4. Typo: Line 114 “distirbution” → “distribution”

**Ethical Concerns:**

["NO or VERY MINOR ethics concerns only"]

**Final Justification:**

I had initially given 4: borderline accept. In their rebuttal, the authors were able to answer my queries and also further expanded on the inverse scaling mechanism. I still have slight apprehensions about the novelty of this work, but I think the results presented so far in this paper are useful and interesting. Therefore, I now recommend this paper as an "5: Accept" to this conference.

**Limitations:**

Authors have adequately discussed limitations of their work.

**Quality:**

3

**Strengths And Weaknesses:**

## Strengths

1. The paper is well-written with clear figures.
2. The authors have done a comprehensive analysis, exploring multiple aspects like temperature effects, loss scaling patterns, and game-specific variations.
3. I particularly find the inverse-scaling analysis to be very interesting and the overall analysis carried out in Section 7.
4. I also appreciate that authors have released their code for reproducibility of their experiments.

## Weaknesses

1. To me, overall it feels like:
- Jones (2021): Already showed scaling laws exist in board games
- Michaud et al. (2023): Already provided general theory connecting Zipf's law to scaling laws
- This paper: Applies Michaud's theory to Jones's domain.

Therefore, the novelty feels limited. The authors essentially took two existing pieces (Jones's empirical observation + Michaud's theory) and combined them. But since Michaud's theory was already modality-agnostic, this feels more like validation than discovery.

2. While the empirical observations are interesting, it's unclear how these findings can improve RL algorithms in general. The empirical validation is very narrow - only discrete, tree-structured board games. There's no evidence this applies to partial observability, or other RL environments.

3. The inverse scaling explanation, while interesting, remains largely speculative despite the experiments in Section 7.3. I feel in order to make this a strong accept, I would want more in-depth analysis of “selective forgetting” explanation provided on lines 275-284.

---

> ### Author Rebuttal · Authors · 2025-07-30
>
> Thank you for your review. We appreciate that you find our paper clear and well-written with a comprehensive analysis of the problem. We address the points you brought up in order:
>
> 1) While we do build on the works of Jones [1] and Michaud et al. [2], we disagree that our paper only amounts to validation. Michaud's quantization model is only modality-agnostic within supervised learning, and does not apply to RL directly (our paper would have been much shorter if it did). The various experiments we perform probe how far does the quantization model apply to AlphaZero training, and where this model fails to apply, e.g. going from loss scaling to Elo scaling.
> Additionally, section 7 (inverse scaling) does not deal with the quantization model, though it is inspired by it.
>
> 2) We are glad you find our observations interesting. The scope of our experiments is indeed limited to fully-observable board games, similar to Jones [1]. One could probably expand this analysis by using algorithms tailored to games with hidden information and stochastic dynamics [3,4]. However, we feel that the amount of experiments already makes this paper quite packed in its current form.
> Secondly, we note that RL training of autoregressive LLMs, a highly active research area, can also be seen as a discreet, tree-structured game with open information, albeit a non-MARL setting.
>
> 3) We are glad you liked our explanation on inverse scaling. Following your suggestion, we added a new appendix section explaining in detail the selective forgetting mechanism we suggest in section 7. We appended the new section to the end of this comment.
>
> ### Questions:
>
> 1) The surprising findings are the loss scaling (section 6) and inverse scaling results (section 7), which already amount to half the paper. Finding Zipf's law in AlphaZero board games is novel, but not surprising, given the results of Blasius & Tönjes [5], as we clearly state in section 5.
>
> 2) That's an interesting point. It's impossible to guess in advance what game will have clearly-identifieable learned task quanta, and as we discuss in the paper, finding even a few such quanta in games like chess is quite hard. We believe the best candidate would be some artificial game built similarly to Michaud et al.'s artificial "multitask sparse parity" dataset. Perhaps a game that involves solving instances of the multitask sparse parity problem in order to gain advantage in the game.
>
> 3) We believe part of the reason why large models overfit on late-game states is exactly what you suggest, i.e. that these states have lower variance in their outcome. That is the point we were trying to make in lines 275-284. We hope the new appendix section added below will answer your question.
>
> 4) Thanks for noticing, we fixed the typo.
>
> We hope we have addressed all your concerns, and encourage you to look at other new sections we added to the paper found in our replies to reviewers c7VB, a847 and Vr29. If we did, we hope you would consider updating your score. Otherwise, we would be glad to answer any remaining questions.
>
> ### References
>
> [1] Jones "Scaling scaling laws with board games." (2021).
>
> [2] Michaud et al. "The quantization model of neural scaling." (2023).
>
> [3] Perolat et al. "Mastering the game of Stratego with model-free multiagent reinforcement learning." (2022).
>
> [4] Cui et al. "Adversarial diversity in hanabi." (2023).
>
> [5] Blasius and Tönjes. "Zipf’s law in the popularity distribution of chess openings." (2009).
>
>
>
> # New appendix section: Inverse scaling mechanism
>
> In section 7.3, we mention a hypothetical mechanism that could cause the observed inverse scaling phenomenon. Here we discuss this mechanism in more detail.
>
> We make the following main assumptions:
> - Agents fit states in descending order of frequency. An agent with a neural network of size $N$ can perfectly fit the values of $n=N/c$ states, where $c$ is the neural capacity needed to fit a single state (Michaud et al. 2023).
> - As the agent improves during training, game outcomes (i.e. state values) change. The values of early-game states change significantly during training, while the values of late-game states change only slightly, if at all.
> - The shift of state values causes neural plasticity loss due to dormant neurons (Sokar et al. 2023).
>
>
> The first assumption is based on the quantization model of neural scaling (Michaud et al. 2023).
>
> The second assumption is based on the fact that, in most board games, player skill greatly influences the value of early-game states. In the games discussed in this paper, a match between random agents will usually end in a draw ($v=0$) while a match between optimal agents will end in a first-player victory ($v=\pm1$), except for checkers. In contrast, late game states that are played a few steps before the game ends have values that are largely independent of player skill, due to power imbalance. Very late states have fixed values if MCTS can probe the entire game tree stemming from them.
>
> The third assumption is based on the work of Sokar et al. (2023), who showed that changing the labels of a dataset during training causes some neuron activations to die out. These so-called "dormant neurons" effectively reduce the neural capacity of the model.
>
> Assume we train a small agent with $10^m$ neurons and a large agent with $10^k$ neurons, $k>m$. In our case, the small agent roughly fits $10^m/c \approx n_{early}$ states, where $n_{early}$ are the number of early-game states with higher frequency than late game states. In Oware and Checkers, $n_{early}$ falls in the range $[100,200]$, see Fig. 5. The large agent fits $10^k/c \gg n_{early}$ and mostly memorizes the values of late-game states.
>
> For simplicity, let us treat the change of state values as a single, abrupt event during training. After this event, both models have high loss on early-game states, but maintain the same loss as before on late-game states, since the latter group kept their value labels. Due to plasticity loss, both models can now fit fewer states than before. The small agent now has high loss, since it could only fit the first $n_{early}$ states before values changed, meaning most of the states it memorized now have new labels. The large model, on the other hand, maintains a lower loss, since it could fit orders of magnitude more states than the first $n_{early}$ states, most of which are late-game states.
>
> It is likely that the small agent, which now has loss close to that of a random agent, will continue to train and re-learn as many states as it can, starting from the most frequent ones. Eventually it will regain low loss on most or all of the first $n_{early}$ states. In contrast, the large agent maintains low loss, and could be close to a local minimum of the loss landscape that fits late-game states well but assigns wrong labels to early-game states. To leave this minimum and re-learn the first $n_{early}$ states, the large agent will have to sacrifice knowledge of some late-game states, since it lost a fraction of its neural capacity. In this scenario, the large model could remain stuck in the local loss minimum and never regain performance on early-game states.
>
> In reality, value labels shift gradually rather than abruptly, meaning that loss will increase gradually during training rather than spike abruptly. We observe this loss degradation in Fig. 7, where large agent value loss on early-game states starts to degrade mid-training.

---

> > ### Comment · Reviewer_hdGQ · 2025-08-04
> > **Response to Authors**
> >
> > Thank you authors for your detailed rebuttal and for adding the explanation for Inverse Scaling Mechanism. I still have slight apprehensions regarding the novelty of this work, but I think that in its current form, this paper presents interesting results and therefore I have now raised my score to an "accept".

---

### Official Review · Reviewer_c7VB · 2025-07-03

**Clarity:** 3
**Significance:** 2
**Originality:** 2
**Rating:** 4
**Confidence:** 3

**Summary:**

The authors study how neural‐scaling phenomena occur in AlphaZero agents on four board games (Connect Four, Pentago, Oware, Checkers). They show that the frequency of board states follows Zipf's law, which arises from the tree structure of legal moves in these games.
The agents reduce value-head loss on states in descending frequency order, which aligns with the “quantization” theory of LLM scaling.
Two of the games (Oware, Checkers) exhibit inverse scaling, beyond a size threshold the Elo falls because high-capacity nets overfit to strategically trivial late-game positions, which gives an anomalous bump in the Zipf curve.

**Questions:**

- How large is "exceedingly large amounts of of data" to visualise Zipf curve skewness in checkers?
- What does "strongly fixed labels" mean at the end of page 8?
- Why is it surprising that the large models have better loss on late game states? Wouldn't we expect this given late game states have lower variance?

Minor:
Page 2 Line 72: change end of equation (2) to be a comma, and Where to where.

**Ethical Concerns:**

["NO or VERY MINOR ethics concerns only"]

**Final Justification:**

I have increased my score to 4, I think this paper explores interesting aspects of games and provides useful analysis, therefore can be accepted at the conference. I still have reservations that the exploration is only done on small 2D deterministic games, so the significance may be limited.

**Limitations:**

This is discussed in the paper clearly, in page 9.

**Quality:**

3

**Strengths And Weaknesses:**

Stengths:
- This paper highlights an interesting pattern for four games with branching states, and draws the connection with Zipf's law.
- The paper is in general well-written and clear.
- There is a good range of empirical analyses on the four games.

Weaknesses:
- The authors make an assertion "It is easy to show that an ideal game, with a constant branching factor and a tree structure free of loops, would produce a Zipf-like distribution", I think it would improve the paper to actually show this, e.g. in an Appendix.
- Some of the discussion of results does not quite align. For example, it is claimed that ``larger agents achieve better loss'' in the caption of Figure 3. However, we see that the yellow curve, which has the highest parameters, is actually in the middle of A (train set) and not the lowest for B (ground truth).
- The connection between RL games and LLMs seem a bit tenous. E.g. the worse performance for larger models is just part of the convergence issues of nonlinear function approximators (c.f. deadly triad), there is no need to mention LLM training (page 5).
- The validation of this theory is on fairly small 2-D games, it is uncertain how this would extend for more complicated games or those having a stochastic transition.
- The error bars only appear occasionally, e.g for the time taken in Figure 3C. How might these look for 3A or B?

---

> ### Author Rebuttal · Authors · 2025-07-30
>
> Thank you for your review. We are glad that you find our paper interesting and well-written, with a good range of empirical analyses. We address the points you brought up in order:
>
>
> - We thank you and reviewer cnDZ for this suggestion, we agree that showing explicitly why this claim is true would improve the paper. We added a new appendix section verifying our claim, appended at the end of this comment.
>
> - Thank you for pointing this out, we now added a footnote to the paper clarifying why this happens. The reason why the largest agent seems a bit worse than the second-largest agent on ground-truth Connect Four is due to a combination of noise and approaching the optimal-play limit.
> The agents we use were taken from Neumann & Gros's work [1]. If you look at their figure 2, you will see that the largest Connect Four agents all have roughly the same performance, because they are already close to the level of an optimal player. For completeness, we decided not to exclude the largest agents in our analysis, even though their performance already stagnates at the largest model sizes.
> One would therefore expect that the largest agents have similar loss values, which is the case. The small difference in loss between the largest agents is likely due to noise.
> To clarify this issue, we added the following footnote to figure 3:
> "The largest nets approach the optimal play limit [5], causing loss to stagnate as model size increases to the largest values."
>
> - We mention the quantization model of Michaud et al. [2] in page 5 since our paper discusses its potential application to AlphaZero training. We mentioned LLMs in page 5 even though the quantization model applies more generally, since Michaud et al. assume Zipf's law originates from the modality, i.e. natural language. However, we now changed the wording of the paragraph header ("Comparison to LLMs" --> "Comparison to the quantization model") and changed the text to dicuss this model in general, rather than LLMs.
> Regarding your other point on worse performance of larger models, discussed in section 7.3: this is indeed an RL phenomenon not seen in LLMs, and we state that explicitly in the text.
>
> - Indeed, similar to prior works (Jones [3], Neumann & Gros [1]), our paper explores deterministic 2D games that are easier to tackle than games like chess with a larger state space. While additional experiments on a wider range of problems could strengthen our results, we feel the paper is already packed with results in its current form, and leave such experiments for future work.
>
> - Figures 3A and 3B are already a bit cluttered, which is why we did not add error margins that could reduce visibility. However, we do see their necessity, especially since they can help clarify the issue you brought up earlier with figure 3. We will add error margins to figures 3A and 3B in the camera-ready version.
>
>
> ### Questions:
>
> - We say "exceedingly large amounts of data" since this is a log-log plot, and extending it requires increasing the dataset size by orders of magnitude. We cannot know inadvance the exact dataset size that's sufficient to visualize the checkers distribution skewness. The required dataset size should be larger by some unknown number of orders of magnitude. Notice that although figure 4C goes up to rank $10^7$, the dataset used to generate it is much larger. First, because we cut off the majority of states from the graph (roughly 90% of states) since they only appeared once or twice during training. Second, because each state with frequency $n$ appears in the dataset $n$ times.
>
> - "Strongly fixed labels" means labels that change very little, if at all, during training. End-game states tend to have such labels, since by that point in the game the outcome is almost determined and even a random agent is almost guaranteed to produce the same outcome as a perfect player. We added a new appendix section expanding on this, which you can find at the end of our reply to reviewer hdGQ.
>
> - What's surprising is that larger agents have a clear preference to reducing loss on end-game states, considering that they have worse performance on early-game states. As you suggest, one would expect all agents to have lower loss on end-game states due to low variance, but that applies to small agents as well.
>
> - Thank you for noticing the typo, we fixed it.
>
>
> Other than the new addition below, we encourage you to look at other new sections we added to the paper, which you can find in our replies to reviewers hdGQ, a847 and Vr29.
> We hope we have addressed all your concerns. If we did, we hope you would consider updating your score. Otherwise, we would be glad to answer any remaining questions.
>
> ### References:
>
> [1] Neumann and Gros. "Scaling laws for a multi-agent reinforcement learning model."  (2022).
>
> [2] Michaud et al. "The quantization model of neural scaling." (2023).
>
> [3] Jones "Scaling scaling laws with board games." (2021).
>
>
>
>
>
>
> # New appendix section: Distribution of ideal games
>
> In section 5 we claim that random games played in a toy-model setting will result in a frequency distribution shaped as a series of plateaus, centered around a power law of exponent $\alpha = 1$. We verify this claim here.
>
> ## Frequency distribution function
> Consider a random game setting following the rules presented in section 5, namely:
>
> - Each game lasts $K$ turns.
> - Each turn $t$, a move $a_t$ is sampled uniformly from $b$ options.
> - Each board position $s$ can only be reached by a single sequence of moves $\{ a_1, ... a_t\}_s$.
>
> The position at turn $t$ is effectively sampled uniformly from $b^t$ possible positions, since the $t$ moves $a_i$, $i \in \{1, \dots, t\}$  that lead to it are each sampled uniformly from $b$ options.
> Since $K$ turns are played each game, when sampling a board position randomly from all games played the frequency of sampling board state $s$ is:
>
> $P(s) = P(t(s)) \cdot P( s | t=t(s))=\frac{1}{K} \cdot \frac{1}{b^t} $ ,
>
> where $t(s)$ is the turn number of board state $s$ and $P(t(s))$ is the probability to sample from turn $t(s)$.
>
> Sorting the board positions in descending order of frequency results in a series of plateaus, each corresponding to a turn number. The number of board states in plateau number $t$ is $b^t$. Plateau number $t$ starts after board position number $n_{start}(t)$, defined by:
>
> $n_{start}(t) = \sum_{i=1}^{t-1} b^i = \frac{b^{t} - b}{b - 1} $,
>
> since each preceding plateau $i$ contains $b^i$ states.
> Board state number $n$ belongs to plateau number $t$ for the integer $t\in \mathbb{N}$ that satisfies:
>
> $n_{start}(t) \leq n < n_{start}(t+1) $,
>
> or:
>
> $\frac{b^{t} - b}{b - 1} \leq n < \frac{b^{t+1} - b}{b - 1} $.
>
> Shifting terms in the inequality and taking the log we get:
>
> $b^t - b \leq (b-1)n <  b^{t+1} - b $
>
> $ b^t  \leq (b-1)n +b <  b^{t+1}  $
>
> $t \cdot log(b)  \leq log [ (b-1)n +b ] <  (t+1)log(b)  $
>
> $t  \leq \frac{log [ (b-1)n +b ]}{log b}  <  t+1$
>
> since $t \in \mathbb{N}$, this is equivalent to using the floor operator:
>
> $    t(n) = \left\lfloor \frac{\log \left[ (b-1)n +b \right]}{\log b} \right\rfloor $. (21)
>
> Using Eq. 13 one sees that the frequency of the $n$'th most common position, $s_n$, is proportional to:
>
> $    P(s_n) = \frac{1}{K} \cdot \frac{1}{b^{t(n)}} $,
>
> where $t(n)$ is defined in Eq. 21. This is the series of plateaus seen in Fig. 2A.
>
>
> ## Distribution is bounded around power law
> Let us define $\widetilde{t}(n) \in \mathbb{R}$ as $t(n)$ without the floor operator:
>
> $   \widetilde{t}(n) =  \frac{\log \left[ (b-1)n +b \right]}{\log b}  $ .
>
> Since $x - 1 < \left\lfloor x\right\rfloor \leq x$, $P(s_n)$ is bounded by:
>
> $    \frac{1}{K} \frac{1}{b^{\widetilde{t}(n)}} \leq P(s_n) < \frac{1}{K} \frac{1}{b^{\widetilde{t}(n)- 1}} $.
>
> Expanding the bounding functions we get:
>
> $    \frac{1}{b^{\widetilde{t}(n)}} = \frac{1}{b^{\frac{\log \left[ (b-1)n +b \right]}{\log b}} } = \frac{1}{e^{\log \left[ (b-1)n +b \right]}} = \frac{1}{ (b-1)n +b} $,
>
> leaving us with:
>
> $    \frac{1}{K} \frac{1}{ (b-1)n +b} \leq P(s_n) < \frac{1}{K} \frac{b}{ (b-1)n +b} $ .
>
> We therefore see that the frequency distribution $P(s_n)$ is bounded by two straight lines (power laws with power 1) with different coefficients, which appear as two parallel lines in log-log scale, as we observe in Fig. 2A.

---

> > ### Comment · Reviewer_c7VB · 2025-08-05
> >
> > Thank you for the authors for the responses to my initial concerns and questions. I think the suggested response would enhance the paper.
> >
> > If Figures 3A and 3B are too clustered after adding the error margins, I think also another figure in the Appendix for the interested reader would serve equally well (with appropriate reference to it in the main paper).
> >
> > ### Frequency distribution function
> >
> > Thank you for agreeing to add an additional frequency distribution example, I have a comments.
> > I think that it should be $a_1, \dots, a_{t(s)}$. In this assumption, do you mean that there is only a single sequence of moves of length $t(s)$ that can be reached per board position? Hence this sort of ideal game works for connect four, but not for something more complicated like chess?
> >
> > I think of $s$ as a particular board state. Let us take $S$ to be the random variable denoting the (random) board state. (Else you should make it clear that $s$ is a random variable.) Then what you want is that
> >
> > $$
> > P(S = s) = P(S = s \text{ and } t = t(s)) = P(t = t(s)) \cdot P(S = s \mid t = t(s)) = \frac{1}{K} \cdot \frac{1}{b^{t(s)}}.
> > $$
> >
> > Every single position would be uniquely reached by one particular (sub)sequence. And under this assumption the earlier states will be more likely simply because of the branching number of possibilities $b^{t(s)}$ is less. Your series of plateaus are simply plotting out the possible states and observing that the probabilities will be the same if $t(s) = t(r)$ for two states $s=r$.
> >
> > If you can tidy up the typos and just make the wording a bit clearer, I think this would be a good example.

---

> ### Author Response · Authors · 2025-08-05
> **Reply to reviewer c7VB**
>
> Thank you for your reply, we are glad you like the new additions to the paper.
>
> Regarding your comments on the new frequency distribution section:
>
> - Thank you for pointing out the typos (t --> t(s), s --> S when random variable), we have now fixed them.
> -  "In this assumption, do you mean that there is only a single sequence of moves of length $t(s)$ that can be reached per board position? Hence this sort of ideal game works for connect four, but not for something more complicated like chess?" : Yes, that is what we meant. However note that Connect Four, like chess, does not behave like an ideal game. Connect Four states can have many possible move sequences leading to them, e.g. playing the moves 1,2,3,4 vs. 3,4,2,1 (the number is the column number where the piece is placed). Chess is indeed more complicated, but in this respect the two games are similar.
> - "Your series of plateaus are simply plotting out the possible states and observing that the probabilities will be the same if $t(s) = t(r)$ for two states $s=r$.": thank you for pointing this out. We originally did not include the full mathematical derivation because we thought this simple explanation is sufficient. We realize now we did not state it clearly in the text. We now added a version of this explanation to the start of the new appendix section:
>          "In short, in a simple branching game the number of possible states at turn $t$ increases as a power of $t$, and all states at turn $t$ have an equal probability to appear  in a game, hence the series of frequency plateaus observed in Fig. 2A".
> - To make the wording a bit clearer, we changed the 3rd bullet point in our definition of an ideal random game: "Each board position $s$ played at turn $t(s)$ can only be reached by a single, unique sequence of moves $\\{ a_1, ... a_t(s) \\}_s$. This means that  one can define a state $s$ only by the sequence of moves played to reach it."

---

> > ### Comment · Reviewer_c7VB · 2025-08-06
> >
> > I think there are still a couple minor points to sort out:
> > > In short, in a simple branching game the number of possible states at turn $t$ increases as a power of $t$.
> >
> > This is a bit confusing as it can be interpreted as either $t$ is the base or exponent. I would suggest "the number of possible states is exponential in the turn number $t$".
> >
> > I don't really understand the notation:
> > $ \{a_1, ... a_t(s) \}_s$.
> > At the very least the $(s)$ needs to belong to the subscript $t(s)$. Is your outer subscript indicating that this is a sequence of moves associated with position $s$?

---

> > > ### Author Response · Authors · 2025-08-06
> > >
> > > Thank you for spotting it, we fixed the phrasing of that sentence according to your suggestion.
> > >
> > > Regarding the term $ \\{ a_1, ... a_t(s)\\}_s $, yes, the outer subscript denotes that this sequence of moves is unique to state $s$. The term $a_t(s)$ is not displayed correctly in Markdown, what we wrote is  `a_{t(s)}`, as you suggested.
> > >
> > > Perhaps a better phrasing is:
> > > "... can only be reached by a single, unique sequence of moves $ \boldsymbol{a}_s = \(a_1, a_2, ..., a_t(s) \) $."
> > > Again, Markdown can't display`a_{t(s)}` correctly, the math term is `\boldsymbol{a}_s = \(a_1, a_2, ..., a_{t(s)} \)`. We will appreciate any other suggestion you have for this term that improves clarity.

---

> > > > ### Comment · Reviewer_c7VB · 2025-08-08
> > > >
> > > > Thank you, I think the new example looks good. Happy to revise my score with the proposed changes.

---

### Note · Authors · 2025-08-11

We would like to thank the reviewers once again for the constructive rebuttal. Their comments helped us write several new additions that we agree have improved the paper significantly.

Since our discussion with reviewer Vr29 was cut short, we would like to clarify that we will revise the camera ready version by taking option 1 (moving appendix C, "Temperature and correlation with scaling laws", to the main section), as well as adding the comment we promised to the main section. We would of course be willing to change the section order in another way if the reviewers find it important enough, and pass the information through the AC.

---

### Decision · Program_Chairs · 2025-09-17

**Decision:**

Accept (spotlight)

**Comment:**

The paper investigates neural scaling in AlphaZero agents across four board games, Pentago, Connect Four, Oware and Checkers. They observe that the frequency of board states follows Zipf's law due to the tree structure of legal moves. They make a connection to the quantization theory of LLM scaling, and they show agents reduce loss on states in descending frequency order. The analysis includes a good range of empirical studies, exploring aspects like temperature effects and loss scaling patterns. A particularly interesting finding is the inverse scaling observed in Oware and Checkers, where high-capacity nets overfit to strategically trivial late-game positions, leading to an anomalous bump in the Zipf curve and a drop in Elo score beyond a certain size threshold. The authors have released their code for reproducibility.

This is a strong paper due to its novel application of Zipf's law to game states, its empirical rigor, and its connection to broader theories of neural network scaling. The identification of inverse scaling in specific game contexts is a significant contribution, highlighting complexities in scaling laws for RL.

The reviewers have highlighted some important minor directions of improvement, and the authors have suggested many concrete changes, including multiple new appendices. I’m sure it goes without saying, but please implement these changes for the camera ready version.